behaviour/ecology/molecular biology

zooarchaeology, livestock, Iron Age, biomolecular archaeology, ZooMS, anthropogenic impacts

# Collagen fingerprinting traces the introduction of caprines to island Eastern Africa

Courtney Culley[1,2], Anneke Janzen[2,3], Samantha Brown[2,4], Mary E. Prendergast[5], Jesse Wolfhagen[2], Bourhane Abderemane[6], Abdallah K. Ali[7], Othman Haji[7], Mark C. Horton[8], Ceri Shipton[9,10], Jillian Swift[2,11], Tabibou A. Tabibou[12], Henry T. Wright[13], Nicole Boivin[1,2,14,15] and Alison Crowther[1,2]

[1]School of Social Science, The University of Queensland, St Lucia, Queensland, Australia
[2]Department of Archaeology, Max Planck Institute for the Science of Human History, Jena, Germany
[3]Department of Anthropology, The University of Tennessee, Knoxville, USA
[4]Institute for Scientific Archaeology, University of Tübingen, Tübingen, Germany
[5]Department of Anthropology, Rice University, Houston, TX, USA
[6]Centre National de Documentation et de Recherche Scientifique, Mutsamudu, Anjouan, Comoros
[7]Department of Museums and Antiquities, Zanzibar, Tanzania
[8]Cultural Heritage Institute, Royal Agricultural University, Cirencester, England
[9]Institute of Archaeology, Gordon Square, University College London, London, UK
[10]Centre of Excellence for Australian Biodiversity and Heritage, College of Asia and the Pacific, Australian National University, Canberra, Australia
[11]Department of Anthropology, Bernice Pauahi Bishop Museum, Honolulu, HI, USA
[12]Centre National de Documentation et de Recherche Scientifique, Moroni, Grand Comore, Comoros
[13]Museum of Anthropological Archaeology, University of Michigan, Ann Arbor, Michigan, USA
[14]Department of Anthropology, National Museum of Natural History, Smithsonian Institution, Washington, DC, USA
[15]Department of Anthropology and Archaeology, University of Calgary, Calgary, Canada

CC, 0000-0002-1485-6999; AJ, 0000-0002-6371-7469;
SB, 0000-0001-5001-525X; MEP, 0000-0003-0275-6795;
JW, 0000-0002-1354-4870; JS, 0000-0002-7436-1947;
NB, 0000-0002-7783-4199; AC, 0000-0002-2394-1917

The human colonization of eastern Africa's near- and offshore islands was accompanied by the translocation of several domestic, wild and commensal fauna, many of which had long-term impacts on local environments. To better understand the timing and nature of the introduction of domesticated caprines (sheep and goat) to these islands, this study applied collagen peptide fingerprinting (Zooarchaeology by Mass Spectrometry or ZooMS) to archaeological remains from eight

**Author for correspondence:**
Courtney Culley
e-mail: courtney.culley@uq.net.au

Iron Age sites, dating between *ca* 300 and 1000 CE, in the Zanzibar, Mafia and Comoros archipelagos. Where previous zooarchaeological analyses had identified caprine remains at four of these sites, this study identified goat at seven sites and sheep at three, demonstrating that caprines were more widespread than previously known. The ZooMS results support an introduction of goats to island eastern Africa from at least the seventh century CE, while sheep in our sample arrived one–two centuries later. Goats may have been preferred because, as browsers, they were better adapted to the islands' environments. The results allow for a more accurate understanding of early caprine husbandry in the study region and provide a critical archaeological baseline for examining the potential long-term impacts of translocated fauna on island ecologies.

# 1. Introduction

Situated at the nexus of ancient exchange and migration routes that connected mainland Africa with the Indian Ocean world, island eastern Africa is a key region for understanding past human impacts on insular ecologies (e.g. [1–7]). Recent archaeological research has examined the introduction of various African and Asian taxa to the islands through human migration and maritime exchange, as well as their potential ecological impacts [8–10]. Human impacts on marine systems and terrestrial resources through long-term subsistence and trade practices have also been investigated [1,11–21]. These transformations appear to date from when the islands were first permanently occupied by food-producing groups during the Iron Age (*ca* 100–1000 CE) (e.g. [1,2,17,22–31]).

Less attention, however, has been paid to questions surrounding the introduction of livestock such as cattle (*Bos taurus* and Zebu, *B. indicus*) and caprines (goat, *Capra hircus* and sheep, *Ovis aries*) to the islands. This is despite the fact that goats have been introduced to more islands worldwide than any other mammal apart from cats (*Felis catus*) and rats (*Rattus rattus*), often with negative environmental consequences including erosion, de-vegetation and overgrazing [32]. Reconstructing caprine dispersals in coastal eastern Africa is complicated by the high rate of bone fragmentation in archaeological sites [1,2,33,34] and the close morphological similarity between sheep and goat bones [35–38]. These issues often render their bones indistinguishable from one another, and from other similar-sized wild bovids (e.g. [39,40]). To address this issue, this paper uses the biomolecular method of collagen fingerprinting to identify caprines to species level at eight archaeological sites across island eastern Africa. The main aim of this study is to refine our understanding of local faunal introductions in their broader socioeconomic contexts and to provide critical baseline information for assessing longer term processes of anthropogenic island transformation and change. The results have enabled us to refine the chronology of caprine introductions to the islands, showing that goats were present from the seventh century CE and sheep by the eighth century CE, the latter some three centuries earlier than previously thought.

# 2. Background

## 2.1. Island colonization and species translocations

Scattered along 3000 km of coastline between Somalia and Mozambique, insular eastern Africa comprises approximately 200 islands in the Lamu, Zanzibar, Mafia, Kilwa, Comoros archipelagos and Madagascar (figure 1). The islands range in size from large continental landmasses (e.g. Madagascar, greater than 580 000 km$^2$) to moderately sized islands (e.g. Unguja, greater than 1600 km$^2$ and Pemba, approx. 1000 km$^2$) and smaller islets, many of which are still uninhabited today. They also have vastly different geologies and biogeographic histories, creating diverse environments and ecologies that would have influenced aspects of human migrations, including economic decisions about which wild or domestic species to introduce to support long-term habitation. For example, some islands such as Unguja, Mafia and Kilwa are relatively low-lying, dominated by sparse shrubland thickets that overly thin soils and coral rag. Others, such as Pemba and Ndzuani, have more variable topography, higher levels of rainfall and/or natural water sources, deeper soils and more substantial forests. The native biota also varies greatly between islands, despite in some cases their relatively close proximity to one another. For example, Pemba, which has been separated from the African mainland for approximately 5 million years by a deep marine channel [41], and Comoros, which are oceanic volcanic islands that

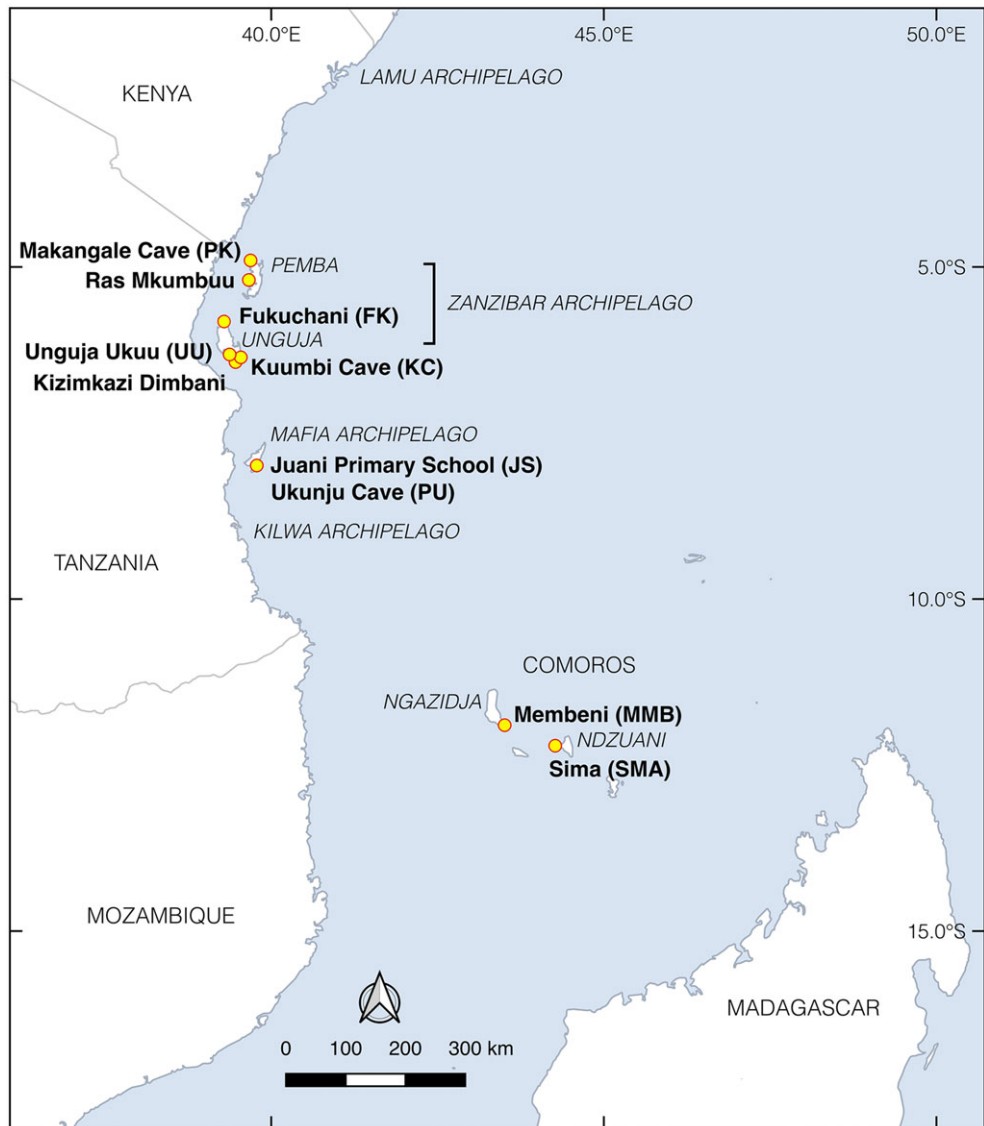

**Figure 1.** Map of island eastern Africa and sites mentioned in the text.

have never been connected to the mainland, are relatively depauperate in terrestrial mammals compared to Unguja and the near shore islands in the Lamu, Mafia and Kilwa archipelagos [21].

The earliest archaeological evidence of human settlement of insular eastern Africa comes from the Later Stone Age, when—from as early as *ca* 20 000 years ago—groups practising a foraging and hunting economy occupied Unguja. The island was still connected to the mainland at this point and appears to have been abandoned when sea levels rose *ca* 9000 years ago [1,31,42,43]. Claims for the presence of foragers on Madagascar from at least 4000 years ago [25,44], and later on Mafia [45] and the Comoros [46], are still debated (e.g. [47,48]). Archaeological research has firmly demonstrated the widespread settlement of the near shore islands by ironworking, ceramic using communities during either the Early (*ca* 100–650 CE) or Middle Iron Age (*ca* 650–1000 CE), a process linked to the spread of agriculture through sub-Saharan Africa [27,49]. Upon settling the coast and islands, these communities became increasingly integrated in long-distance maritime trade networks, particularly during the Middle Iron Age when inter-regional interaction intensified, also bringing traders and migrants from the Middle East, South and Southeast Asia to the region [17,29,50]. Local foraging populations also persisted after the arrival of food production, when various caves sites appear to have been (re)occupied during the Iron Age, perhaps to formulate new exchange networks [31,51]. Small agricultural trading villages appear for the first time on northeast Madagascar and Comoros from around 800–1000 CE [3,52], though these islands have more complex settlement histories involving migrations from both Africa and Southeast Asia, the chronological relationship between which are still poorly resolved [17,53,54].

The eastern African Iron Age was thus a dynamic period of cultural interaction and mobility, which led to the introduction of various wild, domestic and commensal species (some highly invasive) to the mainland coast and islands [21,55]. These included domesticated crops such as various African millets (*Sorghum bicolor*, *Pennisetum glaucum* and *Eleusine coracana*) and Asian rice (*Oryza sativa*) [17,27,56] and fauna such as caprines, cattle (including South Asian zebu), cat, chicken (*Gallus gallus*) and black rat, among others (e.g. [9,57,58]). Human settlement of the islands was also accompanied by the movement of various wild fauna, both from the mainland and between the islands [21,59], further shaping biodiversity across the region. For example, bush pig (*Potamochoerus larvatus*) and blue duiker (*Philantomba monticola*), both of which are commonly found archaeologically on the mainland coast [60,61], were transferred to Unguja, Pemba, Mafia, the Comoros and Madagascar, while common tenrec (*Tenrec ecaudatus*) and two species of lemur (the common brown lemur, *Eulemur fulvus* and the mongoose lemur, *Eulemur mongoz*), were introduced to Comoros from Madagascar [21,59,62]. While the chronology of some of these translocations remains unclear [21,63], archaeological evidence suggests they were introduced from at least the Middle Iron Age onwards [59].

Many endemic insular fauna also faced extirpations and extinctions in the face of human colonization. The extinction of Madagascar's megafauna is perhaps one of the better studied (though still widely debated) cases (e.g. [64–69]), but recent archaeological evidence has also implicated humans in the extirpation of a suite of wild fauna on Unguja [1]. Species that were adapted to a range of open and closed environments, including zebra (*Equus quagga*), reedbuck (*Redunca redunca*) and bush duiker (*Sylvicapra grimmia*), were present in zooarchaeological assemblages from at least the late Pleistocene (*ca* 20 000 years ago) until *ca* 700–1000 CE, but are not found on the island today. While environmental isolation following Unguja's separation from mainland Africa likely contributed, new hunting pressures and habitat impacts associated with the arrival of farming populations (e.g. the clearance of natural vegetation for settlement and agriculture and the introduction of commensal species such as black rat) are also thought to have had a significant impact [5,9,70,71].

## 2.2. Tracing the introduction of caprines to insular Eastern Africa

While the pattern and chronology of wild and commensal faunal introductions to island eastern Africa have begun to be scrutinized [1,9], large-scale comparative work on livestock and particularly caprine introductions is lacking. Archaeological remains of goat, sheep and cattle have been recorded at numerous sites in the eastern African interior from at least 5000 years ago [72–74], where they were an integral part of mainland Pastoral Neolithic economies. They first appear in zooarchaeological assemblages on the mainland coast and hinterland during the Middle Iron Age [75,76], where all three taxa have now been identified using Zooarchaeology by Mass Spectrometry (ZooMS) in coastal Kenya [77]. Current zooarchaeological evidence suggests they were introduced to the islands at about the same time [2,30,33,34,78–81].

The earliest zooarchaeological evidence of goat on the islands dates from *ca* 700 CE [22–25, 30,34,59,79,82]. Sheep, on the other hand, have only been identified at three sites in the Zanzibar archipelago dating to *ca* 1000–1200 CE [20,33], suggesting a much later and perhaps less widespread introduction compared to goats. Zooarchaeological evidence suggests that sheep only arrived in the Comoros at around the same time [3,22]. The earliest evidence for caprines on Madagascar dates from *ca* 900 to 1000 CE [83–85]. Although islands in the Mafia archipelago were colonized as early as *ca* 300 CE by groups with Early Iron Age material culture (including ceramics), there is no robust evidence as yet of either domesticated plants or animals in their economy until the Middle Iron Age, though preservation issues may be partly to blame [86]. Also problematic are claims for the introduction of goat as well as other domesticates (dog, *Canis familiaris* and chicken) to the islands between 5000 and 3000 years ago [79,87], which have not been backed up with robust chronological or osteological evidence [88] or replicated by subsequent studies [1,2,27,31].

A key issue with tracing the arrival of sheep and goat on eastern Africa's coast and islands is the lack of positively identified remains from archaeological sites. The zooarchaeological identification of caprines is complicated by the high frequency of poorly preserved faunal assemblages, as well as the limited number of skeletal elements (such as the metatarsal and astragalus) that have sufficient diagnostic markers for differentiating sheep and goat [37,38]. Without positive taxonomic identifications, the role of caprines in early island economies remains unclear, particularly as hunting wild fauna persisted alongside fishing and shellfish collection as key components of subsistence even after the introduction of food production during the Iron Age [2,11,34]. Our ability to accurately reconstruct past practices such as herd compositions and management strategies, as well as the

preferential introduction of goat versus sheep to the islands, is also constrained. These distinctions are significant as sheep and goat are behaviourally different with characteristic dietary preferences and ecological impacts [40,89].

To overcome these taxonomic limitations, recent studies have applied biomolecular approaches such as proteomics and ZooMS to identify caprines in African assemblages [39,40,77]. ZooMS is a minimally invasive method that measures collagen, typically from bone, using soft ionization mass spectrometry. This provides peptide markers, or 'fingerprints', that often allow for the identification of taxa using predetermined unique markers with particular mass-to-charge ratios ($m/z$) [90]. Here, we apply ZooMS to further our understanding of the timing and nature of sheep and goat introductions to eastern Africa's islands.

# 3. Methods

## 3.1. Sites

Eight sites in the Zanzibar, Mafia and Comoros archipelagos were selected for ZooMS analysis (table 1): Makangale Cave on Pemba, and Fukuchani, Unguja Ukuu and Kuumbi Cave on Unguja, all in the Zanzibar archipelago [1,2,31]; Juani Primary School and Ukunju Cave on Juani in the Mafia archipelago [47]; and Membeni and Sima on Ngazidja and Ndzuani, respectively, in the Comoros archipelago (see electronic supplementary material, data S1 for detailed site descriptions). The sites vary in size, complexity and chronology, with Juani Primary School being the only Early Iron Age assemblage sampled. All other assemblages in the study date from the Middle Iron Age. Makangale and Kuumbi Caves have intermittent evidence of occupation, possibly by forager groups that coexisted with early food-producing communities. Zooarchaeological and archaeobotanical records from these sites indicate a preference for hunting wild fauna while crops are extremely rare [1,17,27,91]. By contrast, village sites including Fukuchani, Membeni, Sima and Unguja Ukuu comprised small- to medium-sized settlements with mainly daub dwellings (though stone structures are also present at Unguja Ukuu) [92,93]. These were occupied by communities with mixed subsistence economies that included hunting, fishing and crop cultivation [2,11,27,86].

## 3.2. Sample selection

The sites analysed in this study have over 6000 individually catalogued bones from Iron Age contexts that, prior to this research, were morphologically identified (i.e. on the basis of comparative anatomy using locally relevant reference collections) to different degrees of taxonomic specificity (see [1,2,9,27,47,86]). Due to high rates of fragmentation, bones were often only able to be identified morphologically to broader taxonomic categories (e.g. 'vertebrate', 'mammal' and 'bovid') and body size class (adapting [94] to eastern African bovids). For example, 'Bovid Size 1' is the size of dik-dik or suni, 'Bovid Size 1–2' is the size of bush duiker, klipspringer, oribi or small caprine and 'Bovid Size 2' is the size of caprine, bushbuck or Thomson's gazelle (see also electronic supplementary material, table S1.1 in data S1).

Given that the main aim of the present study was to use ZooMS to test the presence of caprines, a targeted sampling strategy was employed, focusing on fragmented bones that were identified morphologically as caprine (e.g. '*Capra hircus*' and 'caprine') or less identifiable specimens that could potentially be caprine (e.g. 'Bovid Size 1–2' and 'Bovid Size 2'). For three sites (Unguja Ukuu, Fukuchani and Juani Primary School), we also sampled a random selection of highly fragmented specimens that could only be identified to broader taxonomic levels (e.g. 'mammal' or 'vertebrate'), in order to assess the usefulness of ZooMS in identifying domesticates among poorly preserved bones. This was particularly important at sites where, based on morphology alone, caprines were either rare (Fukuchani) or absent (Juani Primary School). In total, 394 individual faunal specimens were selected from eight sites for ZooMS analysis.

## 3.3. ZooMS protocol

Between approximately 10 and 80 mg of bone was removed per specimen for ZooMS analysis. For broken bones, samples were removed by using sterilized pliers to break off small pieces of bone from already-broken ends ($n = 384$). For unbroken bones ($n = 10$), samples were removed by drilling to

**Table 1.** Summary of results showing the total, bovid-only and sampled Number of Identified Specimens (NISP), the percentage of successful samples per site, whether ZooMS confirmed (C), improved (I) or failed to improve (F) on the morphological identification, and the earliest dates associated with ZooMS-identified caprine remains.

| archipelago | island | site[a] | site type[b] | total NISP[c] | bovid NISP[d] | no. ZooMS samples | % successful ZooMS samples[f] | sheep I | sheep C | goat I | goat C | caprine C | caprine F | earliest associated date[h] |
|---|---|---|---|---|---|---|---|---|---|---|---|---|---|---|
| Zanzibar | Pemba | PK | C | 1315 | 5 | 11 | 100% | | | 3 | | | | goat: 1020–1150 cal CE* |
| | Unguja | FK | 0 | 505 | 169 | 8 | 100% | | | 1 | | | | goat: 7th –9th century CE |
| | Unguja | UU | 0 | 1586 | 455 | 244 | 99% | 9 | | 199 | 14 | 4 | | goat: 655–710 cal CE*; sheep: 685–855 cal CE |
| | | KC | C | 2469 | 595 | 8 | 12% | | | 1 | | | | goat: 530–1475 cal CE* |
| Mafia | Juani | JS | 0 | 94 | 55 | 65 | 6% | | | | | | | |
| | | PU | C | 22 | 2 | 2 | 100% | | | 2 | | | | goat: 7th century CE |
| Comoros | Ngazidja | MMB | 0 | 140 | 72 | 45 | 98% | 1 | | 33 | 3 | | 1 | goat and sheep: 8th–9th century CE |
| | Ndzuani | SMA | 0 | 122 | 17 | 11 | 55% | 1 | | 3 | | | | goat and sheep: 8th century CE |
| totals | | | | 6253 | 1370 | 394 | | 11 | | 242 | 17 | 4 | 1 | |

[a] sites listed in north to south order; see Figure 1 for site codes.

[b] C = cave, 0 = open-air village/port.

[c] total non-human tetrapod NISP, Iron Age levels. At KC, this is phase 1a and 1b only; for all other sites, all contexts are included.

[d] total bovid NISP from Iron Age levels.

[e] references: ([1,2,9,27] Sealinks Project unpublished data [47,86]).

[f] percentage of specimens with successful collagen extraction includes specimens for which species determination could not be made.

[g] C = ZooMS confirmed morphological identification, I = ZooMS improved morphological identification (for example improving from a more general taxonomic identification such as, 'Bovid Size 2' or 'caprine' to a species identification, 'Capra hircus'), and F = due to unsuccessful collagen extraction ZooMS failed to improve the morphological identification.

[h] date ranges marked with an (*) represent associated radiocarbon date(s) on crop seeds, charcoal and/or bone collagen, or OSL dates on ceramics; in all other cases, broad date ranges are provided based on associated ceramics.

remove small fragments while avoiding any morphologically diagnostic features (e.g. epiphyses), to preserve the specimens for future morphological and metric analyses [95]. The samples were separated into two groups for processing: those for which collagen was extracted only for ZooMS ($n = 372$), and those for which collagen was extracted for both ZooMS and stable isotope analyses (the latter results not reported here) ($n = 22$), which required a different extraction protocol.

Bones that were not included in the stable isotope study were analysed using two published collagen extraction protocols [96,97]. Given known concerns about archaeological collagen preservation in the study region [98], we wanted to maximize our ability to identify these samples. Both protocols are acid based; the first, referred to here as the 'acid-insoluble protocol', relies on the demineralization and gelatinization of the bone to extract collagen [90,99], while the second (the acid-soluble protocol) uses the acid-soluble collagen from the initial demineralization [100]. The compatibility of these methods, which can be performed together, allowed us to follow both protocols on the same bone sample and select the best resulting spectra from either protocol for taxonomic identification (external link: doi.org/10.5281/zenodo.4767341).

### 3.3.1. Acid-insoluble protocol

Bone samples were first demineralized in 0.6 M hydrochloric acid (HCl) for at least 18 h, after which the HCl supernatant containing the acid-insoluble fraction was removed and kept aside for the acid-soluble protocol, see below [96,99]. The demineralized bone was rinsed thrice with 50 mM ammonium bicarbonate (AmBic), incubated at 70°C in 100 µl of 50 mM AmBic, and 50 µl of the resulting supernatant was treated with 0.1 µg trypsin (Pierce™ Trypsin Protease, Thermo Scientific) at 37°C for 18 h.

### 3.3.2. Acid-soluble protocol

The supernatant containing the acid-soluble fraction from the initial digestion of the bone (see above) was transferred into 30 kDA molecular weight cut-off ultrafilters and centrifuged at 3700 r.p.m. [97,100]. The sample was rinsed twice with 500 µl of 50 mM AmBic and centrifuged again at 3700 r.p.m. The sample was eluted in 200 µl of 50 mM AmBic, half of which was removed and stored at −20°C as a backup. The remaining 100 µl was then treated with 0.1 µg trypsin (Pierce™ Trypsin Protease, Thermo Scientific) and incubated at 37°C for 18 h.

### 3.3.3. Lyophilized collagen for stable isotope analysis

For the 22 samples that bone collagen was extracted for stable isotope analysis, ZooMS was carried out using the resulting lyophilized collagen from the stable isotope preparation protocol [101,102]. In brief, samples were demineralized in 0.5 M HCl for 1–5 days, rinsed in Milli Q water, treated with pH 3 water and demineralized at 70°C for 24 h. The resulting supernatant was filtered using Ezee Filters and freeze dried in order to lyophilize the collagen. Less than 0.1 µg of the lyophillized collagen was then eluted in 50 mM AmBic and treated with 0.1 µg trypsin (Pierce™ Trypsin Protease, Thermo Scientific) and incubated at 37°C for 18 h.

### 3.3.4. C18 clean-up and MALDI-ToF analysis

Following collagen extraction and digestion for all protocols, the samples were subjected to C18 clean-up with a matrix solution of α-cyano-4-hydroxycinnamic of 10 mg ml$^{-1}$ in 50% ACN/0.1% TFA and allowed to co-crystallize. All samples were spotted onto a ground steel plate in triplicate and analysed using a Bruker Autoflex Speed LRF MALDI-ToF/ToF mass spectrometer. The resulting mass spectra were peak picked with a signal-to-noise ratio of 3.5 using mMass [103] after baseline correction, smoothing and deisotoping with the default parameters. The triplicate spectra for each sample were then averaged using default mMass parameters. Samples were analysed alongside multiple blanks to monitor intra-laboratory contamination, all of which returned negative results and were therefore determined to be empty of collagen.

Diagnostic peptide markers were identified through the comparison of the resulting spectra with an established reference library [98,99,104,105] (table 2; electronic supplementary material, data S2). For specimens to achieve a species-level identification of either *C. hircus* or *O. aries*, all diagnostic species-specific peaks had to be present, whereas those that were missing peaks at COL1α2 757–789 (G) were only able to be identified as 'caprine'. Because wild bovids share many of the same peptide markers,

**Table 2.** ZooMS markers used for taxonomic identifications (see electronic supplementary material, data S2 for full details including published references).

| taxon | COL1a1 507–518 | COL1a2 978–990 | | COL1A2 375[a] | | | COL1a2 484–498 | A2 889 | COL1a2 502–519 | COL1a2 292–309 | COL1a2 793–816 | COL1a2 454–483 | COL1a1 585–617 | | COL1a2 757–789 | |
|---|---|---|---|---|---|---|---|---|---|---|---|---|---|---|---|---|
| *Neotragus moschatus* | 1105 | 1180 | 1196 | 1182(?) | 2056(?) | 2072(?) | 1427 | 1532 | 1580 | 1648 | 2131 | 2792 | 2883 | 2899 | 3017(?) | 3033 |
| *Bos* spp. | 1105 | 1192 | 1208 | — | — | — | 1427 | 1532 | 1580 | 1648 | 2131 | 2792 | 2853 | 2869 | 3017 | 3033 |
| *Ovis aries* | 1105 | 1180 | 1196 | 1154 | 2028 | 2044 | 1427 | — | 1580 | 1648 | 2131 | 2792 | 2883 | 2899 | 3017(?) | 3033 |
| *Capra hircus* | 1105 | 1180 | 1196 | 1154 | 2028 | 2044 | 1427 | — | 1580 | 1648 | 2131 | 2792 | 2883 | 2899 | 3077 | 3093 |
| *Cephalophus*[b] | 1105 | 1192 | 1208 | 1154 | 2028 | 2044 | 1427 | 1574 | 1580 | 1648 | 2131 | 2792 | 2853 | 2869 | 3043 | 3059 |
| cf. *Philantomba monticola*[c] | 1105 | 1192 | 1208 | 1154 | 2028 | 2044 | 1427 | 1560(?) | 1580 | 1648 | 2131 | 2792 | 2883 | 2899 | 3043 | 3059 |
| Canidae | 1105 | 1210 | 1226 | — | — | — | 1453 | — | 1566 | 1649 | 2131 | 2820 | 2853 | 2869 | 2983 | 2999 |
| Cercopithecinae | 1105 | 1219 | 1235 | — | — | — | 1427 | — | 1580 | 1619 | 2115 | 2832 | 2883 | 2899 | 2958 | 2974 |
| *Eretmochelys imbricata*[d] | | 1220 | | — | — | — | 1443 | — | 1459 | 1572 | | 2790 | 2843 | 2859 | 2899 | 3007 |
| *Dermochelys coriacea*[d] | 1136 | | | — | — | — | 1453 | — | 1459 | 1572 | | | 2854 | 2869 | 2899 | 3007 |
| Felidae | 1105 | 1207 | 1223 | — | — | — | 1453 | — | 1566 | 1609 | 2163 | 2820 | 2853 | 2869 | 2983 | 2999 |
| Suidae | 1105 | 1180 | 1196 | — | — | — | 1427 | — | 1550 | 1648 | 2131 | 2820 | 2883 | 2899 | 3017 | 3033 |

[a] markers in italics (?) are present in the LC-MS/MS but are not suitable for species identification with ZooMS (see [98]).

[b] *Cephalophus adersi* was used as a representative of Cephalophinae, so may represent more than one species (electronic supplementary material, data S2, [98]).

[c] markers were predicted from genetic sequences rather than directly from the LC-MS/MS; therefore, species identification is tentative.

[d] sequences are derived from proteomic sequence data, therefore, species identification is tentative.

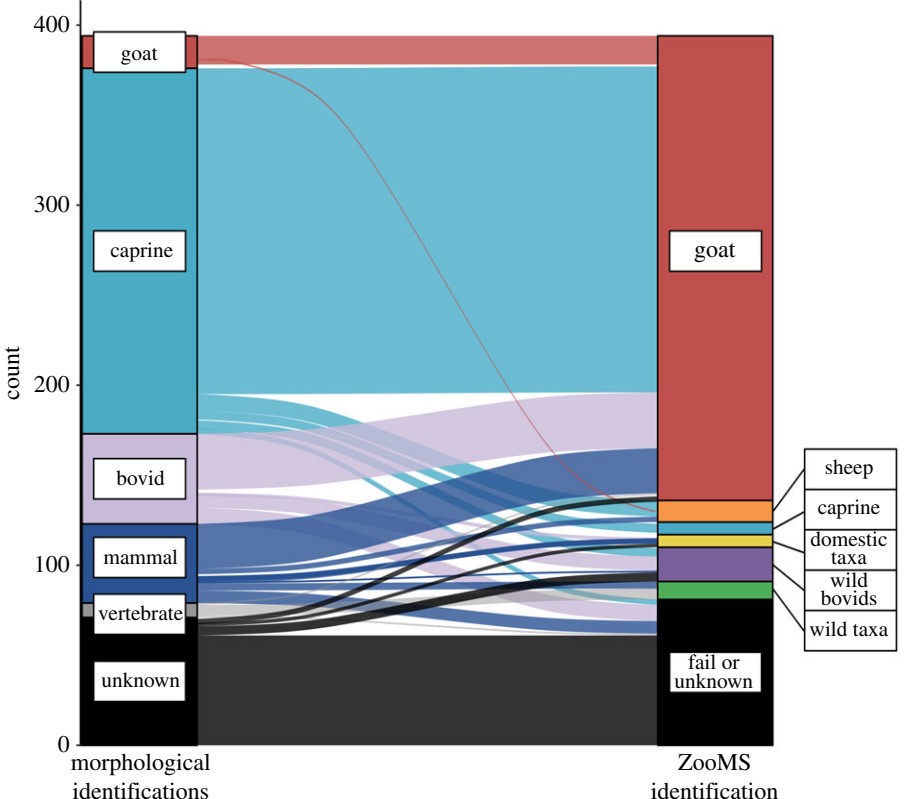

**Figure 2.** Morphological identification of taxa compared with ZooMS identifications. Morphological identifications are ordered from the most specific taxonomic level (goat—top) to the most general one (unknown—bottom); see electronic supplementary material, data S2 for details.

some specimens were only able to be identified to coarser taxonomic groups [98]. Samples containing high amounts of collagen but where no identification was possible when comparing the peaks to the reference library are reported here as 'unknowns', which may be identifiable in future with improvements to the reference dataset.

# 4. Results

The overall success rate of collagen extraction was extremely high at 81% ($n = 318/394$) (table 1; see electronic supplementary material, data S2 for full list of results). Of the samples analysed using the published acid-based ZooMS protocols, all but 10 performed better (i.e. lower baselines and a greater number of diagnostic peaks which matched ZooMS reference library spectra overall) using the acid-insoluble protocol, in which the bone was demineralized and gelatinized. These results are consistent with a recently published study which showed that gelatinization of bone was the optimal collagen extraction method for samples from sites with known preservation issues [106]. All 22 samples extracted from lyophilized collagen from the stable isotope preparation protocol produced identifiable spectra (electronic supplementary material, data S2). Despite the high success rate overall, collagen yields varied considerably between sites. The best preservation rate was at Unguja Ukuu on Unguja (99%, $n = 242/244$ samples), while the worst was at the Juani Primary School site on Juani ($n = 4/65$, 6%) (table 1). Variations in collagen preservation between sites likely relate to local microenvironmental conditions such as soil composition and pH [107–109], and are being investigated further.

Of the 318 samples that produced collagen, 314 could be identified to taxon based on the comparison of the ZooMS spectra to the reference library (electronic supplementary material, data S2), leading to significant improvements in taxonomic specificity compared to the morphological identifications (figure 2 and table 1). The majority of the samples were identified using ZooMS as *C. hircus* ($n = 259/318$, 81%). This species was identified at seven of the eight sites investigated (all except Juani Primary School), including for the first time at Makangale Cave, Ukunju Cave and Sima. Seventeen of

**Table 3.** Summary of non-caprine ZooMS identifications.

| family/taxon | subfamily | ZooMS identification | PK | FK | UU | JS | MMB |
|---|---|---|---|---|---|---|---|
| Bovidae | Antilopinae | *Neotragus moschatus* (suni) | | 7 | 3 | | |
| | Alcelaphinae | Alcelaphini | | | 1 | | |
| | Antilopinae, Alcelaphinae or Aepycerotinae | Antilopini (oribi), Alcelaphini or *Aepyceros* (impala) | | | 1 | | |
| | Bovinae | *Bos taurus* (cattle) | 1 | | | | 3 |
| | Bovinae or Antilopinae | *Bos* spp., *Syncerus caffer* (buffalo) or *Sylvicapra* spp. (duiker) | | | | 1 | |
| | Cephalophinae | cf. *Philantomba monticola* (blue duiker) | | | 4 | 1 | |
| | Cephalophinae | *Cephalophus* spp. | | | | 2 | |
| Canidae | | Canidae (dog) | | | 2 | | |
| Cercopithecidae | | Cercopithecinae (monkey) | | | 2 | | |
| Cheloniidae | | cf. *Eretmochelys imbricata* (hawksbill turtle) | 4 | | | | |
| Cheloniidae/ Dermochelyidae | | cf. *Dermochelys coriacea* (leatherback turtle) or *Eretmochelys imbricata* (hawksbill turtle) | 2 | | | | |
| Cetacea | | whale/dolphin/porpoise | | | 1 | | |
| Felidae | | *Felis* spp. (cat) | | | | | 1 |
| Suidae | | Suidae (pig/boar) | | | 1 | | |

these 259 ZooMS identifications confirmed previous *C. hircus* morphological identifications (table 1), whereas the remainder had previously been identified as caprine ($n = 181$), bovids identifiable only to size class but not to any narrower taxonomic grouping ($n = 31$), mammals ($n = 26$), vertebrates ($n = 1$) or indeterminate ($n = 3$), identifiable only to size class (see electronic supplementary material, data S2). A small number of *O. aries* ($n = 11/318$, 3%) were identified at three sites (Unguja Ukuu, Membeni and Sima), all for the first time. These were previously morphologically identified as Mammal Size 2 ($n = 2$) and caprine ($n = 9$).

Other domestic or commensal species identified in our sample included *Bos* spp. (cattle) ($n = 4/318$, 1%) at Membeni and Makangale Cave, Canidae ($n = 2/318$, less than 1%) at Unguja Ukuu, and *Felis* spp. (cat) ($n = 1/318$, less than 1%) at Membeni (table 3). Canidae are most likely to be the domestic dog, *Canis familiaris*, as no other species from the same genera (e.g. black-backed jackals, *C. mesomelas*) are native to or known to have been introduced to these islands in the past. The introduced commensal wildcat (*Felis silvestris lybica*), along with domestic dogs, have been identified previously at the sites on morphological grounds [2,20] and via ancient DNA analysis [10].

Our ZooMS analysis also identified a range of wild fauna in the assemblages including various bovids (e.g. *Neotragus moschatus* [suni], *Cephalophus* spp. [duikers] and cf. blue duiker), suids, Cercopithecinae (Old World monkeys), Cheloniidae (either leatherback sea turtle, *Dermochelys coriacea*, or hawksbill sea turtle *Eretmochelys imbricata*), along with a specimen that is probably cetacean (marine mammal) (table 3). The turtle specimens matched closely to recently published turtle markers [105]; however, work on reptiles is still developing and needs further elaboration to provide certainty for these identifications. Four specimens that yielded collagen did not match any published ZooMS reference spectra and were therefore unable to be taxonomically identified.

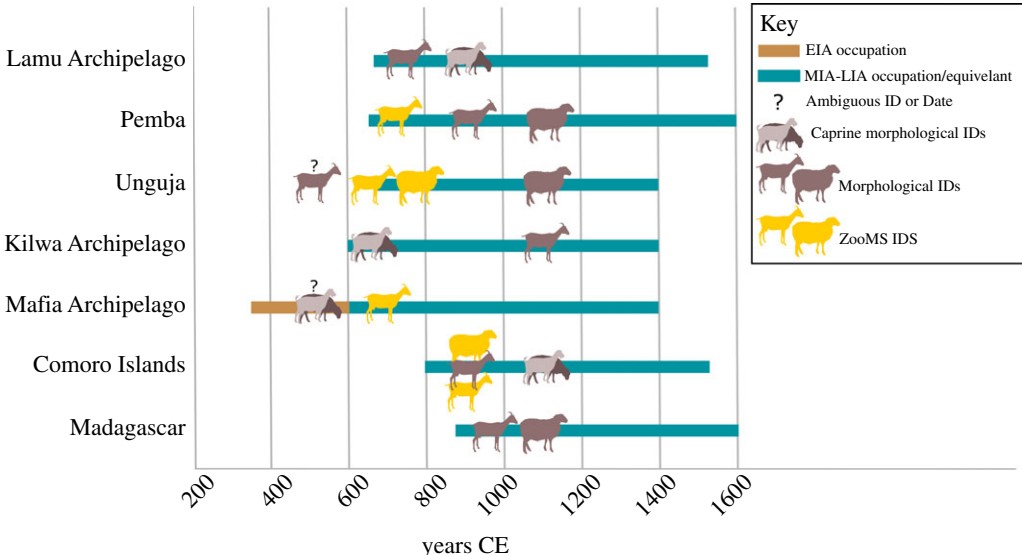

**Figure 3.** Chronology of earliest archaeological caprine (*Ovis/Capra*), sheep (*Ovis aries*) and goat (*Capra hircus*) remains in island eastern Africa, comparing morphological and ZooMS identifications. Chronologically ambiguous goat remains from Unguja Island and Mafia archipelago pre-date established occupation dates for these islands and are therefore questioned. See detailed site summary and references in the electronic supplementary material, table S1.2 in data S1.

## 5. Discussion

### 5.1. Diachronic patterns in the introduction of caprines

The broad chronology of caprine introductions to eastern Africa's islands can now be reconstructed, drawing on associated radiocarbon dates and/or ceramic chronologies (table 1; electronic supplementary material, table S1.2 and S1.3 in data S1). These data show a number of interesting patterns of which two stand out: first, all ZooMS-identified caprines are associated with seventh- to tenth-century archaeological contexts. No caprines were identified in the Early Iron Age assemblage from the Juani Primary School site, although as discussed further below, this may be a result of sample size and taphonomic factors affecting collagen preservation at the site. These dates align with previous zooarchaeological evidence showing patterns of livestock introductions to the mainland coast beginning in the Middle Iron Age, around the time of intensifying local and global trading networks [1,2,9,27,31,86]. Second, goats were not only much more abundant overall, but appear to have arrived on the near shore islands around one or two centuries before sheep (figure 3). The presence of sheep on the islands in the late first millennium CE is of further significance given nearly all previous findings based on morphological identifications had indicated that sheep were only introduced in the early second millennium CE [3,33,34].

Based on the associated chronological evidence, the earliest goat specimen in our study come from the large trading port of Unguja Ukuu, Zanzibar. This specimen originates from the lowermost excavated layers of the site (i.e. the oldest) dating to 655–710 cal CE (Trench 11) and 670–770 cal CE (Trench 14) based on Bayesian modelling of associated radiocarbon dates ([17,27]; see electronic supplementary material, table S1.3 in data S1 for further details). Dates for the appearance of goat at other Zanzibar sites, including Fukuchani (595–880 cal CE) [17,27] and Kuumbi Cave (530–1475 CE) [31], are based on bracketed radiocarbon and OSL ages, respectively, which give much lower resolution time ranges. Likewise, the earliest occurrence of goat on Pemba (at Makangale Cave), and in the Mafia archipelago (in the lowermost excavated layers of Ukunju Cave), can only be broadly dated to seventh–tenth century CE based on associated Middle Iron Age Triangular Incised Ware pottery (e.g. [110]).

Sheep do not appear on the islands until approximately a century later than goat, with the earliest occurrence being at Unguja Ukuu in a layer dating just prior to 685–855 cal CE (see electronic supplementary material, data S1) [27]. Both sheep and goat appear together in Comoros in several of the earliest layers at Membeni and Sima in association with eighth- to tenth-century pottery [17]. Before this study, the earliest occurrence of sheep on the near shore islands was from Ras Mkumbuu on Pemba (post-eleventh century) [33] and Kizimkazi Dimbani on Unguja (twelfth century) [20], though they had been recorded on Madagascar in sites dating from the ninth and tenth centuries [83,84].

The fact that both caprine species make their first appearance on the islands at Unguja Ukuu, which is one of the region's oldest trading ports [30,111–113], raises the question of whether caprines were introduced to eastern Africa through maritime trade in addition to the well-documented terrestrial route through the mainland interior [114]. This idea is worth considering in the light of the fact that at least one other large domesticated bovid, the zebu, was likely introduced to eastern Africa from South Asia through Indian Ocean trade [114–116]. Maritime routes of introduction have also been proposed for goat in other regions of the Indian Ocean [117,118], lending further support to this hypothesis. However, until further genetic work can be undertaken on modern and/or ancient goats to assess phylogeographic relationships (for review see [119]), the most parsimonious explanation is that caprines were introduced to the islands from the mainland. The comparatively early presence of caprines at Unguja Ukuu most likely relates to sampling bias (the site had the largest assemblage analysed in this study and also has the most robust radiocarbon chronology, allowing more precise time frames for caprine arrivals to be established) as well as its larger size and wealth as a major Indian Ocean trading hub compared to other sites in the study. The latter probably afforded its occupants better access to domesticates than more rural localities such as neighbouring Fukuchani. These questions nonetheless warrant further investigation in order to better understand the role of trade—both its economic and social dimensions—in the spread and/or acquisition of domesticates by different communities during this period, and the implications for the spread of caprines to other near- and offshore islands.

The absence of caprines in the Early Iron Age assemblage from the Juani Primary School site presents a major zooarchaeological problem for establishing baselines for livestock arrivals to the islands (and indeed, for the Early Iron Age in general; [27]). As noted above, this site had the poorest collagen survival rate in this study, at just 6% ($n = 4/65$). The absence of domesticates in the Early Iron Age levels is therefore difficult to interpret and may be as much a preservation issue as a genuine absence. It nonetheless highlights that there is still no unequivocal evidence for Early Iron Age domesticates (either plant or animal) anywhere in island eastern Africa. The ZooMS results thereby support previous zooarchaeological research at the site that showed the importance of hunting alongside maritime resources such as shellfish and fish to Early Iron Age economies, particularly during island colonization [86].

The identification of a single goat specimen at Kuumbi Cave, which has a long history of hunter–gather occupation [1,31,120], is an important find in relation to debates about pre-Iron Age introductions of domesticates to eastern Africa's islands. Previous research [42,79] had morphologically identified a range of domestic taxa at the site, including caprines, cattle, chicken, dog and cat, dating from as early as 5000 years ago. However, subsequent zooarchaeological and chronometric analyses at the site failed to replicate these findings [121], identifying only wild taxa in all occupation phases [1,31]. The goat specimen identified in this study confirms the presence of domesticates at the site, at least in the Middle Iron Age. It also shows that later forager communities kept livestock or acquired meat, presumably from neighbouring agro-pastoral communities, as also observed at contemporaneous mainland sites [77].

## 5.2. Island herd compositions

The ZooMS results suggest that goat had much greater economic importance in early island economies compared to sheep—a pattern that had been previously suggested based on morphological identifications [2,20,33,34,75]. Not only do sheep appear much later than goat on the islands, they are also much rarer in zooarchaeological assemblages. Across all sites sampled here, sheep comprise just 12 specimens identified by ZooMS (representing a minimum of seven individuals), compared to 258 specimens for goat (representing a minimum of 56 individuals) (electronic supplementary material, table S1.4 in data S1). Sheep were also less ubiquitous overall, being present at just three sites compared to seven. This apparent preference for goat may have been for environmental reasons, or may reflect cultural factors such as culinary practices or symbolic values [122], or secondary resource exploitation [123]. Alternatively, because sheep are more specialized to graze rather than browse [35,123], they may have been considered less suited to herding in the islands' environments, which tend to be dominated by forests and coral thicket scrub rather than pastures suitable for grazing. Even today, goats are far more common on Zanzibar than sheep, comprising 99.2% of caprines kept by households [124]. Goats are highly adaptable and resilient domesticates and their relatively small size, at least compared to other livestock like cattle, make them easier to transport between the islands. They are often preferred over sheep by herding communities where cattle are the dominant livestock

because cattle and sheep compete for the same fodder, especially if the available grazing grounds are limited [125–129].

## 5.3. Wild faunal extirpations and translocations

The identification of wild fauna at the sites through ZooMS sheds further light on broader questions relating to faunal extirpations and translocations. One unexpected finding, for example, was the presence of an alcelaphine bovid (a tribe including topi, hartebeest and wildebeest) on Unguja, given this taxon has not previously been identified in archaeological assemblages on any of eastern Africa's islands and is not extant on any today. The specimen was identified in a layer at Unguja Ukuu dating to 715–855 cal CE. The environmental requirements of alcelaphines include a preference for open habitats and a diet based primarily on grasses, which are inconsistent with present-day small browsing bovids found on the island. One possibility is that the species was part of a relict population that, like other larger-bodied, primarily grazing bovids identified at Kuumbi Cave (e.g. zebra, buffalo and waterbuck), survived on the island after it became separated from the mainland until eventually becoming extirpated [1]. Alternatively, its presence at Unguja Ukuu may be due to the translocation of either whole or parts of animal carcasses to the site from the mainland. Of note, the ZooMS-identified element was a rib, which is a commonly transported part of the carcass.

Additionally, the presence of duikers (Cephalophinae) on Juani Island in the Mafia archipelago may be evidence of anthropogenic pressures on small island ecosystems. These species are not found on Juani today, though neighbouring islands still support small browsing bovids such as the suni [130]. Presently, members of the genus *Cephalophus* are widespread on the islands, as is the blue duiker, which is found widely throughout Africa including most eastern African islands except Juani [131]. The two cephalophine bovids and the cf. blue duiker samples from Juani Primary School all originate from the Early Iron Age layers, which raises the possibility that they were either translocated by early settlers from nearby Mafia island (where blue duiker are still found today) or were naturally occurring on the island during the Early Iron Age and were later extirpated.

## 5.4. Long-term ecological impacts of caprines on Eastern Africa's islands

For the first time, this study has provided a robust archaeological baseline from which we can begin to study the potential ecological effects of caprines on eastern Africa's islands. Archaeological benchmarks such as these are increasingly recognized as being critical for the current management and restoration of island ecosystems [7,70,132]. An abundance of global research has demonstrated the devastating effects that caprines and other introduced livestock can have on insular environments. Goats are generalist browsers that thrive in almost any environment [40,114,133,134], whereas sheep are more selective grazers with less adaptive fluidity [35]. Goats can impact environments directly through browsing, grazing and trampling as well as indirectly by modifying vegetation, leading to a decline in native biomes (plant and animal communities) and altered nutrient cycles [32,135–137]. This can often lead to other invasive species also dominating ecosystems. Still poorly understood, for example, is how the near-simultaneous introduction of agro-pastoralism alongside invasive animals such as black rat and cat, and other extractive activities such as ironworking may have had interrelated ecological effects over the long term. Although we have very little supporting paleoenvironmental evidence for any of the islands studied here, the vegetation on many today comprises a mosaic of secondary forests, scrubby coastal moorlands, degraded fallow bush, remnants of evergreen lowland coastal forest and agricultural areas [86,138]. In the past, however, it is likely that the coastal forests such as those preserved in small pockets at Jozani on Unguja and Ngezi on Pemba were much more widespread [138,139].

ZooMS data show a clear preference for goats compared to sheep by early island communities in eastern Africa. The widespread introduction of goats across all archipelagos studied during the initial phases of island settlement may have applied new pressures on otherwise vulnerable island ecosystems. More research, particularly on island paleoecology as well as early herding strategies, such as through stable isotopes studies to detect preferences for feeding habitats (e.g. [140,141]), will help shed further light on these issues in future.

# 6. Conclusion

The processes associated with the colonization of eastern Africa's near- and offshore islands have, until now, been largely understood in relation to intensifying Indian Ocean trade, maritime connectivity and

growing social and urban complexity (e.g. [4,26,28,30,58,86,142,143]). Questions concerning subsistence adaptations and longer term ecological transformations, on the other hand, have received much less attention (e.g. [1,2,11,21]), despite global recognition of the importance of archaeology for providing long-term perspectives on island historical ecologies (e.g. [5,144–147]). Drawing on recent methodological advances in biomolecular archaeology, this study used collagen fingerprinting to build, for the first time, a robust empirical baseline for livestock introductions during the early phases of island settlement. It has provided the first clear evidence of the widespread presence of domestic caprines, mainly goats, by the mid–late second millennium CE at sites on all three major archipelagos in the study. Sheep and goat were identified at multiple sites for the first time, showing that their presence was much more widespread and—in the case of sheep—much earlier than previously documented, with dates for the introduction of sheep to the islands pushed back by at least three centuries. The preference for goats compared to sheep likely reflects the fact that, as browsers, goats are much better suited to island environments, signalling the decision-making processes of early farmers as they moved into and adapted to new environments.

On a methodological level, this research has demonstrated the effectiveness of using ZooMS to refine faunal identifications in highly fragmented assemblages from a tropical environment, which is often difficult to achieve owing to poor organic preservation (e.g. [107,148]). While collagen preservation varied widely between sites, overall, it is clear that there was a significant increase in species-specific identifications with ZooMS compared to morphological analyses alone. Application of this technique to a wider range of sites and time periods, as well as the expansion of the reference library, will enable archaeologists to generate more accurate reconstructions of the chronology of livestock introductions, herd compositions and the relationship between hunting and herding as economic strategies, not just in eastern Africa but across the continent more broadly. The archaeological baseline generated in this study can now be used to help understand the potential long-term ecological impacts of faunal translocations on eastern Africa's island environments, particularly when coupled with more palaeoecological and archaeological research on the wider cumulative impacts of settlement, subsistence and trade.

Data accessibility. The datasets supporting this article have been uploaded via an external link: https://doi.org/10.5281/zenodo.4767341.

Authors' contributions. C.C. conceived and designed the study, performed laboratory work and data analysis, and wrote the manuscript. A.J. provided morphological data on the faunal remains, assisted with laboratory work and data analysis, and edited the manuscript. S.B. assisted with laboratory work and data analysis and edited the manuscript. M.E.P. provided morphological data on the faunal remains and edited the manuscript. J.W. provided morphological data on the faunal remains, edited the manuscript and contributed to making figure 2. B.A., A.K.A., O.H., M.C.H., C.S., J.S., T.A.T. and H.T.W. performed fieldwork, contributed archaeological data and edited the manuscript. N.B. conceived the study and edited the manuscript. A.C. conceived and designed the study, performed fieldwork, contributed archaeological data and helped write the manuscript. All authors gave final approval for publication.

Competing interests. The authors have no competing interests.

Funding. This research was funded by the Max Planck Institute for the Science of Human History, The University of Queensland's School of Social Science and a University of Queensland Graduate School (Research Training Program) scholarship awarded to C.C. Fieldwork was funded by a European Research Council grant for the SEALINKS Project (grant no. 206148) awarded to N.B. Radiocarbon dates were funded by UK Natural Environment Research Council grants (grant nos. NF/2012/2/4 and NF/2013/2/1).

Acknowledgements. We are grateful to the Zanzibar Department of Museums and Antiquities, Tanzania's Commission for Science and Technology (COSTECH) and Division of Antiquities (Ministry of Natural Resources and Tourism), and the Centre National de Documentation et de Recherche Scientifique (Comoros) for support of our fieldwork and permission to conduct these analyses. We also thank the staff at the National Museums of Kenya and British Institute for Eastern Africa for their assistance during the sampling process. Permissions to conduct fieldwork were granted by the following authorities: in Tanzania, COSTECH (permit no. 2012-303-ER-2011-85) and Antiquities Division (EA.402/605/01); in Zanzibar, the Office of Chief Government Statistician (Zanzibar Research Committee) and Department of Museums and Antiquities; in Comoros, the Centre National de Documentation et de Recherche Scientifique. We also thank Dr Jana Zech and the laboratory staff at the MPI-SHH for technical support.

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
