## [Peer Review File · Royal Society Open Science]

Review History

RSOS-202341.R0 (Original submission)

Review form: Reviewer 1

Is the manuscript scientifically sound in its present form?

Yes

Are the interpretations and conclusions justified by the results?

Yes

Is the language acceptable?

Yes

Do you have any ethical concerns with this paper?

No

Have you any concerns about statistical analyses in this paper?

No

Recommendation?

Major revision is needed (please make suggestions in comments)

Comments to the Author(s)

Word document attached with comments (see Appendix A).

Review form: Reviewer 2

Is the manuscript scientifically sound in its present form?

Yes

Are the interpretations and conclusions justified by the results?

Yes

Is the language acceptable?

Yes

Do you have any ethical concerns with this paper?

No

Have you any concerns about statistical analyses in this paper?

No

Recommendation?

Accept with minor revision (please list in comments)

Comments to the Author(s)

There are some minor changes needed and some grammatical corrections, otherwise this is an excellent paper.

Page 2 (in the text) line 59 and ff. - It would be worth defining the term Iron Age here and what it is conventionally understood to imply - especially as you use the term frequently in the paper. The use of Early Farming Communities might be better - but much depends on whether you are using the term Iron Age simply as a chronological marker, or whether you also mean to imply inception of farming

Page 3, lines 21-22 - for the benefit of those unfamiliar with the regional archaeological terminology and chrono-stratigraphy, it would be helpful if you could point out that domestic stock had been present in mainland East Africa for much longer associated with Pastoral Neolithic (PN) assemblages - and even in the coastal hinterland by c 3500 BP (Wright 2005 on Tsavo sites, although he does not provide any morphological identifications, just sized classed inferences owing to the presence of PN ceramics).

Page 5, line 53 - I prefer Late Stone Age rather than Later Stone Age, as it is grammatically more correct (Early, Middle and Late). 'Later' has become the default in southern African archaeology, but there are still East Africanists who use 'Late' - and this was more common in earlier decades. Please consider changing.

Page 7 (Methods, section 2.1) Lines 16-17 - I'd like to see some acknowledgement that all of these sites had been located prior to the Sealinks work - important though the contributions of that project have been. How about this phrasing here (or something akin to it): 'Located during previous archaeological campaigns led by different local and foreign researchers, these were re-visited and re-excavated between ...'

Same page line 32 - stone structures were also present at Unguja Ukuu - see Fitton, T. and Wynne-Jones, S., 2017. Understanding the layout of early coastal settlement at Unguja Ukuu, Zanzibar. *Antiquity*, 91(359), pp.1268-1284.

Perhaps acknowledge that Chami exposed structural evidence for houses at Juani Primary School Chami, F.A. 2000. Further archaeological research on Mafia Island. *Azania* 35: 212-13, and maybe also make reference to more recent geoarchaeological work aimed at detecting house floors (can you not use 'hut' please! refer to them as daub or mud-and-wattle houses) Sulas, F., Kristiansen, S.M. and Wynne-Jones, S., 2019. Soil geochemistry, phytoliths and artefacts from an early Swahili daub house, Unguja Ukuu, Zanzibar. *Journal of Archaeological Science*, 103, pp.32-45.

Page 8 line 3 - replace 'were' with 'was' (a random selection ... was)

Page 12, Fig 3 caption, line 54 - Capital Island for 'Unguja Island' for consistency elsewhere in the paper

Page 13 - again acknowledging the very significant contributions the Sealinks project made to clarifying the settlement and occupation history of Kuumbi Cave, the suggested early presence of SE domesticates and a 'Neolithic' occupation had already been challenged before this phase of work began - especially by Sinclair - perhaps cite his 2007 paper on line 24 after 'finding,' while retaining citations to references 24 & 38 at the end of the sentence. P.J.J. 2007. "What is the archaeological evidence for external trading contacts on the East African coast in the first millennium BC." In *Natural Resources and Cultural Connections of the Red Sea*, edited by J. Starkey, P. Starkey and T.J. Wilkinson, 1-8. Oxford: Archaeopress.

Page 14, line 7 - reference to UU being one of the oldest trading ports on the Swahili coast merits a reference to the ongoing work by Wynne-Jones, Sulas and Fitton at the site - Wynne-Jones, Sulas & Fitton

Wynne-Jones, S., Sulas, F., Out, W.A., Kristiansen, S.M., Fitton, T., Ali, A.K. and Olsen, J., 2021. Urban Chronology at a Human Scale on the Coast of East Africa in the 1st Millennium ad. *Journal of Field Archaeology*, 46(1), pp.21-35.

Page 16 - line 52 - an example of a site where this future work on later 2nd millennium faunal assemblages could be carried out is Kua on Juani Island - Christie's work hints at a growing presence of caprines, but again no formal IDs, just size-classed categorisations - see Christie, A.C., 2013. Exploring the social context of maritime exploitation in Tanzania between the 14th-18th c. AD: recent research from the Mafia Archipelago. Prehistoric marine resource use in the Indo-Pacific regions, pp.97-122. Perhaps more information on taxa is available in her 2011 PhD - not sure it is online however.

Page 17 line 7 - change 'subside' to 'subsist' - neither the Cambridge online dictionary nor the online Merriam-Webster use the word 'subside' in the manner implied here - although I have seen it used in this way in other publications.

Review form: Reviewer 3

Is the manuscript scientifically sound in its present form?

Yes

Are the interpretations and conclusions justified by the results?

Yes

Is the language acceptable?

Yes

Do you have any ethical concerns with this paper?

No

Have you any concerns about statistical analyses in this paper?

No

Recommendation?

Accept with minor revision (please list in comments)

Comments to the Author(s)

Review for the manuscript « Collagen fingerprinting traces the introduction of caprines to island eastern Africa”
by Culley et al.

I found the manuscript of Culley and collaborators of particular interest since it focuses on a relatively unexplored area of research: the introduction of domesticates (especially caprines) to islands of eastern Africa. The manuscript describes the application of ZooMS to various archaeological sites aiming at refining morphological identifications of faunal remains. They identified a large number of caprine remains, of which the majority is goat. They discuss the newly identified domesticates site by site and the influence caprines might have had on the islands’ ecologies. Although I found the paper of great interest, I raise in the subsequent paragraphs several points that, according to me, should be improved prior to publication.

Comments:

Generally, when referring to the two caprines species, the term “caprines” is plural. Otherwise, you only refer to goat (by opposition to ovine for sheep). I suggest you check the entire manuscript to avoid misunderstandings.

You make a number of statements that, in my opinion, should be more nuanced and if not, more referenced. For instance, l. 43, you state that sheep and goat distinction is “notoriously difficult” from caprines assemblages. A lot of papers proposed criteria for sheep/goat differentiation (Gillis, Chaix & Vigne, 2011; Salvagno & Albarella 2017 for instance) and you should nuance this sentence by adding something regarding African environments, in which this is particularly the case (and some others).

I found the introduction a bit brief since the context is only quickly described and the method (which, although not broad new, is still developing) is not defined. I also would like a sentence or two describing what you imply when you describe island eastern Africa as “ecologically rich and diverse region” (p. 6, l. 5).

I would also like to know what you mean by “morphological identification”. Do you refer to comparative anatomy only, to geometric morphometrics or a combination of both?

I understood that you used published markers for your species identifications. However, I am highly upset by the fact you mainly refer to an under-review paper for many species ID, which is by essence, not yet accessible. In addition, you did not consider a COL1A1 marker for

sheep/goat distinction in all your data analysis, which appears problematic since it has been highlighted that the “classic” COL1A2 marker is shared between sheep and a large number of African wild bovids and the COL1A1 marker between goat and wild bovids (Le Meillour et al. 2020). I suggest you re-perform your species identification including the COL1A1 marker, to make sure what you identified are indeed (for most of the faunal remains) domestic goat, which I am certain will be the case.

In your discussion, you mention 14C dates. But you do not explain how you obtained them. Was the radiocarbon dating performed on the remains you identified as domesticated caprines or are they related to the burial context (i.e., they were performed on material recovered from the same layer, etc.)?

I do have a problem with your interpretations of herd compositions. I do not criticize the outcomes; but the lack of nuance and most of all, the absence of NMI representation. I understood that you identified mostly goats in your assemblages using ZooMS. However, one bone does not mean one animal. I suggest you rephrase this part taking into consideration the NMI for each site.

Since only a few research teams use ZooMS or palaeoproteomics in African archaeological context, it would be interesting to include in your conclusion a bit of hindsight regarding biomolecular studies concerning caprines herding in Africa. Maybe adding a few sentences in your introduction and including your results into the “larger picture” in the discussion or conclusion would help readers to acknowledge your study.

Finally, I suggest you re-read carefully your text since your sentences are sometimes difficult to follow (mostly due to their length).

p. 5, l. 46: I am not sure that we can still use the term “recently-developed” when referring to ZooMS since the method is employed in archaeology for more than a decade now.

Fig. 1: I am not sure that the right place for this figure is in the introduction, since it appears part b presents some of your results.

p. 8, § 2: why do specify that sheep were identified “morphologically” after you describe the appearance of goats in the previous sentence? Where goats identified using other methods?

p. 9, Sites: I would place Fig. 1 here to locate the sites you are referring to in the text.

p. 10, ZooMS protocol: did you perform MS/MS analyses of your samples? Since you are working on close-related species of bovids and that the presence of wild bovids is highly possible, MS/MS is a strong method to avoid misidentifications.

p. 11, l. 33: when mentioned once, you can refer to the species only by *C. hircus*, *O. aries*, etc. You forgot to put “*hircus*” in italic in the sentence.

p. 12, l. 10: Which species of Alcelaphini tribe did you use in your referential?

Fig. 2 is unclear and difficult to read. I suggest you present the information differently for clarification purposes.

p. 14, l. 12: what taphonomic factors are you referring to?

p. 14, l. 22-23: what do you mean by “confident morphological identifications”?

Fig. 3: I suggest you rephrase the caption which is a bit difficult to follow.

p. 15, l. 3: "the earliest reliably-dated ZooMS-identified goat" formulation is unclear. Do you mean that you dated one of the specimens that presented spectra of domestic goat? Concerning radiocarbon dates, see my comment above. Generally, for your dating paragraph, I suggest you re-order your ideas differently since it is hard to follow and mixing cal. BP, CE and BCE dates.

Same comments for the following paragraphs on sheep.

Decision letter (RSOS-202341.R0)

Dear Miss Culley

On behalf of the Editors, we are pleased to inform you that your Manuscript RSOS-202341 "Collagen fingerprinting traces the introduction of caprines to island eastern Africa" has been accepted for publication in Royal Society Open Science subject to minor revision in accordance with the referees' reports. Please find the referees' comments along with any feedback from the Editors below my signature.

Please submit your revised manuscript and required files (see below) no later than 7 days from today's (ie 09-Mar-2021) date. Note: the ScholarOne system will 'lock' if submission of the revision is attempted 7 or more days after the deadline. If you do not think you will be able to meet this deadline please contact the editorial office immediately.

on behalf of Professor Matthew Collins (Associate Editor) and Pete Smith (Subject Editor)
openscience@royalsociety.org

Reviewer comments to Author:

Reviewer: 1

Comments to the Author(s)

Word document attached with comments.

Reviewer: 2

Comments to the Author(s)

There are some minor changes needed and some grammatical corrections, otherwise this is an excellent paper.

Page 2 (in the text) line 59 and ff. - It would be worth defining the term Iron Age here and what it is conventionally understood to imply - especially as you use the term frequently in the paper. The use of Early Farming Communities might be better - but much depends on whether you are using the term Iron Age simply as a chronological marker, or whether you also mean to imply inception of farming

Page 3, lines 21-22 - for the benefit of those unfamiliar with the regional archaeological terminology and chrono-stratigraphy, it would be helpful if you could point out that domestic stock had been present in mainland East Africa for much longer associated with Pastoral Neolithic (PN) assemblages - and even in the coastal hinterland by c 3500 BP (Wright 2005 on Tsavo sites, although he does not provide any morphological identifications, just sized classed inferences owing to the presence of PN ceramics).

Page 5, line 53 - I prefer Late Stone Age rather than Later Stone Age, as it is grammatically more correct (Early, Middle and Late). 'Later' has become the default in southern African archaeology, but there are still East Africanists who use 'Late' - and this was more common in earlier decades. Please consider changing.

Page 7 (Methods, section 2.1) Lines 16-17 - I'd like to see some acknowledgement that all of these sites had been located prior to the Sealinks work - important though the contributions of that project have been. How about this phrasing here (or something akin to it): 'Located during previous archaeological campaigns led by different local and foreign researchers, these were re-visited and re-excavated between ...'

Same page line 32 - stone structures were also present at Unguja Ukuu - see Fitton, T. and Wynne-Jones, S., 2017. Understanding the layout of early coastal settlement at Unguja Ukuu, Zanzibar. *Antiquity*, 91(359), pp.1268-1284.

Perhaps acknowledge that Chami exposed structural evidence for houses at Juani Primary School Chami, F.A. 2000. Further archaeological research on Mafia Island. *Azania* 35: 212-13, and maybe also make reference to more recent geoarchaeological work aimed at detecting house floors (can you not use 'hut' please! refer to them as daub or mud-and-wattle houses) Sulas, F., Kristiansen, S.M. and Wynne-Jones, S., 2019. Soil geochemistry, phytoliths and artefacts from an early Swahili daub house, Unguja Ukuu, Zanzibar. *Journal of Archaeological Science*, 103, pp.32-45.

Page 8 line 3 - replace 'were' with 'was' (a random selection ... was)

Page 12, Fig 3 caption, line 54 - Capital Island for 'Unguja Island' for consistency elsewhere in the paper

Page 13 - again acknowledging the very significant contributions the Sealinks project made to clarifying the settlement and occupation history of Kuumbi Cave, the suggested early presence of SE domesticates and a 'Neolithic' occupation had already been challenged before this phase of work began - especially by Sinclair - perhaps cite his 2007 paper on line 24 after 'finding,' while

retaining citations to references 24 & 38 at the end of the sentence. P.J.J. 2007. "What is the archaeological evidence for external trading contacts on the East African coast in the first millennium BC." In *Natural Resources and Cultural Connections of the Red Sea*, edited by J. Starkey, P. Starkey and T.J. Wilkinson, 1–8. Oxford: Archaeopress.

Page 14, line 7 - reference to UU being one of the oldest trading ports on the Swahili coast merits a reference to the ongoing work by Wynne-Jones, Sulas and Fitton at the site - Wynne-Jones, Sulas & Fitton

Wynne-Jones, S., Sulas, F., Out, W.A., Kristiansen, S.M., Fitton, T., Ali, A.K. and Olsen, J., 2021. Urban Chronology at a Human Scale on the Coast of East Africa in the 1st Millennium ad. *Journal of Field Archaeology*, 46(1), pp.21-35.

Page 16 - line 52 - an example of a site where this future work on later 2nd millennium faunal assemblages could be carried out is Kua on Juani Island - Christie's work hints at a growing presence of caprines, but again no formal IDs, just size-classed categorisations - see Christie, A.C., 2013. Exploring the social context of maritime exploitation in Tanzania between the 14th-18th c. AD: recent research from the Mafia Archipelago. *Prehistoric marine resource use in the Indo-Pacific regions*, pp.97-122. Perhaps more information on taxa is available in her 2011 PhD - not sure it is online however.

Page 17 line 7 - change 'subside' to 'subsist' - neither the Cambridge online dictionary nor the online Merriam-Webster use the word 'subside' in the manner implied here - although I have seen it used in this way in other publications.

Reviewer: 3

Comments to the Author(s)

Review for the manuscript « Collagen fingerprinting traces the introduction of caprines to island eastern Africa”
by Culley et al.

I found the manuscript of Culley and collaborators of particular interest since it focuses on a relatively unexplored area of research: the introduction of domesticates (especially caprines) to islands of eastern Africa. The manuscript describes the application of ZooMS to various archaeological sites aiming at refining morphological identifications of faunal remains. They identified a large number of caprine remains, of which the majority is goat. They discuss the newly identified domesticates site by site and the influence caprines might have had on the islands' ecologies. Although I found the paper of great interest, I raise in the subsequent paragraphs several points that, according to me, should be improved prior to publication.

Comments:

Generally, when referring to the two caprines species, the term “caprines” is plural. Otherwise, you only refer to goat (by opposition to ovine for sheep). I suggest you check the entire manuscript to avoid misunderstandings.

You make a number of statements that, in my opinion, should be more nuanced and if not, more referenced. For instance, l. 43, you state that sheep and goat distinction is “notoriously difficult” from caprines assemblages. A lot of papers proposed criteria for sheep/goat differentiation (Gillis, Chaix & Vigne, 2011; Salvagno & Albarella 2017 for instance) and you should nuance this sentence by adding something regarding African environments, in which this is particularly the case (and some others).

I found the introduction a bit brief since the context is only quickly described and the method (which, although not broad new, is still developing) is not defined. I also would like a sentence or

two describing what you imply when you describe island eastern Africa as “ecologically rich and diverse region” (p. 6, l. 5).

I would also like to know what you mean by “morphological identification”. Do you refer to comparative anatomy only, to geometric morphometrics or a combination of both?

I understood that you used published markers for your species identifications. However, I am highly upset by the fact you mainly refer to an under-review paper for many species ID, which is by essence, not yet accessible. In addition, you did not consider a COL1A1 marker for sheep/goat distinction in all your data analysis, which appears problematic since it has been highlighted that the “classic” COL1A2 marker is shared between sheep and a large number of African wild bovids and the COL1A1 marker between goat and wild bovids (Le Meillour et al. 2020). I suggest you re-perform your species identification including the COL1A1 marker, to make sure what you identified are indeed (for most of the faunal remains) domestic goat, which I am certain will be the case.

In your discussion, you mention 14C dates. But you do not explain how you obtained them. Was the radiocarbon dating performed on the remains you identified as domesticated caprines or are they related to the burial context (i.e., they were performed on material recovered from the same layer, etc.)?

I do have a problem with your interpretations of herd compositions. I do not criticize the outcomes; but the lack of nuance and most of all, the absence of NMI representation. I understood that you identified mostly goats in your assemblages using ZooMS. However, one bone does not mean one animal. I suggest you rephrase this part taking into consideration the NMI for each site.

Since only a few research teams use ZooMS or palaeoproteomics in African archaeological context, it would be interesting to include in your conclusion a bit of hindsight regarding biomolecular studies concerning caprines herding in Africa. Maybe adding a few sentences in your introduction and including your results into the “larger picture” in the discussion or conclusion would help readers to acknowledge your study.

Finally, I suggest you re-read carefully your text since your sentences are sometimes difficult to follow (mostly due to their length).

p. 5, l. 46: I am not sure that we can still use the term “recently-developed” when referring to ZooMS since the method is employed in archaeology for more than a decade now.

Fig. 1: I am not sure that the right place for this figure is in the introduction, since it appears part b presents some of your results.

p. 8, § 2: why do specify that sheep were identified “morphologically” after you describe the appearance of goats in the previous sentence? Where goats identified using other methods?

p. 9, Sites: I would place Fig. 1 here to locate the sites you are referring to in the text.

p. 10, ZooMS protocol: did you perform MS/MS analyses of your samples? Since you are working on close-related species of bovids and that the presence of wild bovids is highly possible, MS/MS is a strong method to avoid misidentifications.

p. 11, l. 33: when mentioned once, you can refer to the species only by *C. hircus*, *O. aries*, etc. You forgot to put “hircus” in italic in the sentence.

p. 12, l. 10: Which species of Alcelaphini tribe did you use in your referential?

Fig. 2 is unclear and difficult to read. I suggest you present the information differently for clarification purposes.

p. 14, l. 12: what taphonomic factors are you referring to?

p. 14, l. 22-23: what do you mean by “confident morphological identifications”?

Fig. 3: I suggest you rephrase the caption which is a bit difficult to follow.

p. 15, l. 3: “the earliest reliably-dated ZooMS-identified goat” formulation is unclear. Do you mean that you dated one of the specimens that presented spectra of domestic goat? Concerning radiocarbon dates, see my comment above. Generally, for your dating paragraph, I suggest you re-order your ideas differently since it is hard to follow and mixing cal. BP, CE and BCE dates.

Same comments for the following paragraphs on sheep.

===PREPARING YOUR MANUSCRIPT===

===PREPARING YOUR REVISION IN SCHOLARONE===

Author's Response to Decision Letter for (RSOS-202341.R0)

See Appendix B.

Decision letter (RSOS-202341.R1)

Dear Miss Culley,

I am pleased to inform you that your manuscript entitled "Collagen fingerprinting traces the introduction of caprines to island eastern Africa" is now accepted for publication in Royal Society Open Science.

on behalf of Pete Smith (Subject Editor)
openscience@royalsociety.org

Appendix A

Overall, there is a lot of interesting research presented in this paper and a large amount of data contributing to the understanding of animal domestication and translocation in eastern Africa. The authors have done a commendable job of pulling together historic archaeological excavated data sets, radiocarbon dates, morphological info with a robust data set of new collagen fingerprinting analyses. Thus, there are some important insights here which have a wider impact than just presenting the archaeology of these island sites, demonstrated by the paper trying to pull together so many strands from different disciplines. The reach of the paper, however, in the way it is written, is quite narrow in terms of overly focusing on the archaeology of the islands without really broadening this out to the bigger debates on domestication in an East African context, interaction with mainland eastern Africa, as well as how different some of the islands are from each other, and how that might play into a big-picture comparison of the movement of humans, animals, and plants across the different island groups. The text is repetitive, too long, and the language is unclear at points, with long sentences, which makes it hard for the reader to get through to the key points and important new research that has been done. For this work to reach a wide readership in a journal such as Royal Society Open Science, it needs to be much shorter and punchier, without as much specific archaeological information in the main text. Even for a readership of archaeologists working in the region and familiar with sites and the big questions, the arguments are not presented in a clear and organised fashion to follow the take-home messages and significance of the new data presented. At present, it reads much more like a thesis, and thus would benefit from a thorough editing to shorten it and distil the key points for the reader.

The word 'Swahili' is used differently in this paper, sometimes to denote the coast and sometimes to denote the islands as a group, and other times to talk about cultural groups and ways of life. My first point about this is that it needs defining somewhere in the introduction, especially for a broad readership who might associate the word with the language, rather than the cultural or historical meaning or the meaning here to signify a lifeway, heritage, etc. I also wonder, then, about the use of it in the paper to refer to the island groups that are compared in the study— as there are many references, especially in the discussion, to 'Swahili islands'. I wonder if this is a bit problematic to group them in this way, especially without defining the term, and clarifying if this means the time period and way of life that connected the communities living on these islands, or if it is being used as a geographical term? Give this some thought and define it as the authors wish it to be defined and used throughout the paper so that the reader is absolutely clear what is meant by the term.

Abstract

Lines 5-6

This sentence is quite clunky for the abstract. It does not say that this material is from archaeological sites, and the term Iron Age is separated from the chronology, which is difficult for the average reader who might not know that the Iron Age spans that time period in eastern Africa. 'main early colonisation phases' is also quite difficult to interpret

Line 10 - 12

'caprine presence' – odd phrasing – perhaps to be as clear as possible for the abstract, just write that previous zooarch analyses identifying caprines in the remains, or in the faunal record

'this study found..' not very clear - sounds like the team found these remains whilst excavating, or they were lost, rather than identifying these species using collagen fingerprinting, so be clearer here

‘or had only tentatively’ – rather weak point for the abstract, which should be short and punchy and draw the reader in. Remember that some readers only read the abstract, so this should be all of the key points of the paper distilled into these sentences. Is it because the remains were fragmentary, and difficult to identify using standard morphological techniques?

Lines 12 – 23

Too wordy for an abstract. Distil these ideas into one sentence.

Introduction

Best to put a space between the number and the unit ‘3000 km’

For the opening sentences of the introduction, these sentences are not very clear about what this paper is trying to introduce. Trans-continental – yet Africa is a continent and the Indian Ocean world is a rather nebulous entity? Concrete examples here would help as well as being far more plain and clear in the language about the history of exchange between eastern African islands and the Indian Ocean – due to seafaring technology, exchange of specific crops, etc.

What is meant by ‘world regions’?

Explain what is meant by the Swahili coast

The last few sentences of the first paragraph of the Introduction need to be tightened, as it jumps between the work done on the Swahili coast and in a global context on island systems and therefore uses a lot of buzz words – it is then difficult to know what the authors are actually trying to say.

Lines 48-60

‘likely’ is over-used, and is unclear where the introduction of these non-native species came from – how did they get there, and from where?

‘mid-late’ as a classifier for a date range is a bit ambiguous – explain this in numbers for the reader, especially with use of the term Iron Age here again

Lines 9 – 46

‘despite’ used twice in succession. First sentence of this paragraph is quite difficult to understand that caprines are sheep and goat; ‘Western Indian Ocean islands’ ‘islands globally’ – explain?

Line 23

Which islands?

Line 24 - 25

Have not introduced ‘the Swahili’ so difficult for the reader to understand what the ‘Swahili subsistence economies’ means

What does ‘regional’ refer to here?

Lines 42-43

Why is it notoriously difficult? Explain here or combine with above sentence about fragmentation and poorly preserved remains, but also how it is difficult morphologically when you don’t have specific elements of the skeleton, it is compounded by them being fragmentary and poorly preserved. Explain this succinctly for a non-zooarchaeological specialist

Line 47

I don't think collagen fingerprinting is a recent development

Line 51

Delete 'assist in' and be assertive with the findings

1.1 Island colonisation

Section 1.1 could really be tightened up and more helpful to have a comparison of the histories and ecologies of the island groups that are being compared, as they have very different human settlement histories – it would be great to either group/ organise them in this section, or have sub themes to discuss (briefly) where the humans came from and when, and what plants and animals were translocated and whether there was also influence from Indian Ocean trade. Some of this information is in the current section for some of the island groups, but not all. Because we know these histories are all quite different, based on previous work, I think it's important to emphasise this to the reader, especially as the paper groups the islands together for comparison.

Line 14

Delete 'highly'

Line 16

This sentence could usefully be moved to the intro so that the intro covers the interaction between the islands and mainland, but still could be explained much more specifically on what is meant by 'culturally-mediated biological exchanges'. Explain who and where people came from and what they exchanged, what their lifeways and motivations were so that the reader gets a clear picture of what is meant by this phrase.

I find Figure 1B quite hard to read and think it could be better displayed with a horizontal bar perhaps with the date ranges and a clearer comparison between the 2 methods, as it is difficult to see when the species level identifications are lumped with the more general identifications 'mammal' - it is difficult to see that the biomolecular identifications helped attain a better, more specific analysis. Wording of the figure caption is also hard to read : 'caprine related ZooMS identifications' and 'only those samples that returned a biomolecular identification' – what does this mean, in plain language?

Line 11 - 38

'culturally-mediated' is used many times throughout the manuscript without much context to explain who this refers to, what was exchanged, from what directions, why, etc.

Delete 'also'

'recent evidence' – recent archaeological evidence?

'were present archaeologically' – not very clear

'late Pleistocene' – give dates for this in brackets for the reader

Sentence starting 'Their local...' needs to be split up

Line 45

Wild mainland species? As only the bush pig is listed as 'wild' in this sentence

Line 50

'were possibly moved' explain this lack of certainty

Line 57

Colonised by...? Or inhabited by humans? In general, an overuse of the word 'colonised' in the manuscript without a context

Line 9

'is lacking' – better word choice there to end paragraph

Lines 12-34

Explain why 'Iron Age' is in quotes

Would be useful to explain here the use of domestic/wild animals on the mainland, and how this compares with island translocations

'much later' – quantify this if possible

'it is suggested' – based on what?

'Early Iron Age' – first time this is used, without Iron Age ever being contextualised

Why is there no robust evidence? Explain the lack of excavation, etc here

Lines 36-59

'compositions' of what?

Distinctions/distinctive used multiple times in one sentence

'This raises...' what is the chronology of when these events occurred, and is this the case for all of the island groups that are being discussed?

2.1 Sites

Delete 'widescale'

Replace 'antiquity' with chronology or other more precise term

'Early Iron Age' – date?

'Cave sites' sentence too long – also what were these mixed subsistence economies? I don't think by this point the reader has a clear picture of the differences between communities inhabiting the mainland and islands of east Africa in the Iron Age

2.2 Sample selection

'as being as close to caprine as possible' - what does this mean? Explain

'pre-date all confirmed evidence to the Swahili region' – what is this, and when? Is there a Swahili region?

'broader categories' – this is ambiguous

'less specific taxonomic classes' – ambiguous

Only in the final sentence of this section does the reader know that Zooms analysis was used – move this earlier.

2.3 ZooMS protocol

For correct use of 'stable isotope' to describe the analyses say that the bone collagen was also extracted for stable isotope analyses, and that the preparation method to extract collagen for stable isotope ratios or measurements instead of analysing *for* 'stable isotopes' or 'stable isotopes preparation method'. See Chapter 2 of Sharp's book for more help on this:

https://digitalrepository.unm.edu/unm_oer/1/

It is important to explain why you must avoid diagnostic features for future research/ comparing previous analysis

It is also crucial to explain why you use 2 different established ZooMS protocols, as they were developed for specific purposes. The wording here is also a bit strange, as the extraction methods are not necessarily 'accessing' or 'utilising', as the gelatinisation and solubilising during acid demineralisation are not exactly correct here, if strictly following the protocols in the references listed. Explain what buffers you used and at what PH for the extractions, and why it was necessary to follow 2 different methods. These are important to detail, as I also couldn't find a direct comparison in the results and discussion section about the use of the 2 methods. This is important, as one of the issues the results raised was the preservation of collagen from the material and the ability to extract enough collagen for identification. These 2 methods have a big impact on that, as the HCl (depending on molarity, volume, how many times it is changed) will be more aggressive and if not watched carefully and in samples with very little collagen yield already, could destroy the organic. For the readership of Royal Society Open Science, and for archaeologists working on poorly preserved material trying to use these methods, the comparison of these extractions on this large data set is important, and should be highlighted in the Results and Methods, and perhaps include a table or graph in the supplementary showing which method was best, etc. In the Zenodo files, I could only open the spectra in mMass and not necessarily see which extraction method was used for the averaged spectra. I would put the files on Zenodo as text files as well, simply because text files can be imported into any mass spec software programme.

Lines 45 – 56

Need more detail here, specifically about the machine settings, plate used, etc. Refer to the references here and their Methods sections for the details typically added in this section for ZooMS analysis.

Results

I find the first paragraph of the results section difficult to read, as it is essentially a word description of Table 1. Try to summarise all of the results more succinctly, rather than choosing some sites to give detail of the success rate, or phrases such as 'double the success rate' 'very low success rate – worst of all the sites' as this is not really helpful to the reader for understanding a clear report of the results, and doesn't mean much outside of the comparison of this data set. I also think that in cases where there are such different sample sizes (1/8) it is actually a little misleading to try and frame the results as percentages against one another. As this section rightly says, the variations are probably due to a range of different factors, and in the last sentence, it would be helpful to reference the wealth of literature on factors affecting collagen preservation.

Line 28 - what is defined as a 'successful' sample? 'species-specific identification' clunky

Line 33 – hircus in italics

Line 48 – Probable? Explain this.

Line 60 – via ancient DNA? Explain this.

Lines 5-40 on page 10: This paragraph is difficult to read and make sense of, and I wonder if it would be better to be either supplemented or shortened alongside a table instead. Also, some statements need to be explained, such as 'missing too many markers for species identifications'. It is important to have a table in the main text of the paper with the ZooMS markers which were used to identify caprines at the sites, and perhaps a list of the markers which were used to identify the wild species and a simple list with markers of the wild species that were found. As it is, Figure 2 and 3 are helpful to get an overall view, but there is no ZooMS marker data in the paper as a table at present. I realise that there were a lot of samples analysed, so the full data set needs to be in supplementary, but it is important to highlight for non-specialists how the reference markers/spectra were used to make the identifications. This is especially important given that many of the spectra are reported as compared against a data set in a paper that is submitted, but not in press (Janzen et al. submitted).

For the samples that had good spectra, but no identification was possible, it would be useful to explain in the Results section whether the authors tried to match against existing ZooMS data, possible contaminants, or whether the exercise was simply to identify caprines and if not, then it was discounted as 'unknown'

Lines 43-60 - this paragraph is also difficult to read, and could be summarised with a table. Phrases such as 'remained at the caprine level' are unclear. Also, not sure the average reader will understand the various zooarchaeological size classes of "Bovid" or "Mammal Size Class" so this needs to be explained and contextualised if used here. Same problem with caption for Figure 2.

Line 52 - needs a reference here, or explanation of how this is known without LC-MS/MS data

Discussion

For me, the first paragraph and Figure 3 in the Discussion are really key points of the paper, and need to be featured in the abstract and intro as the significant outcomes of the work. Figure 3 really sums up the research nicely and I wonder if incorporating a graphic like this one for Figure 1b would be helpful.

The first paragraph on page 13 has a lot of unclear phrasing 'earliest reliably-dated ZooMS-identified goat occurs' 'here it was found to be present' 'goat first appears in context with a potentially wider timeframe' I'd recommend re-writing this paragraph using clear, succinct language.

Line 35 – explain what you mean by 'lowermost contexts' for a non-specialist

Line 42 – *based on the data presented* –not usual to start a new paragraph with 'on the other hand' as it should follow from thought directly before

Line 45 – 'before now' – *prior to the results presented here?*

Line 51 – 'ZooMS-confirmed sheep occurred' ?

Line 53 – 57 explain what this means to someone who is not an archaeologist reading this

Line 26, page 14 – ‘well-dated’ ‘site’s size and wealth’ – what does this mean – explain this here and give specific details of why these observations are important to the argument

Line 38 – ‘ZooMS-identified caprines’ unclear

Line 55 – ‘It nonetheless highlights’ ... this sentence is key and a really important point, but is couched in between wordy sentences, so would suggest bringing this out and contextualising it as a key point that this paper makes in light of the new data set presented

Line 7, page 15 – ‘maritime-oriented societies’ – what does this mean? Explain.

Line 21 – much greater importance or just now a much greater number? Explain.

Line 36 – lower numbers in your studied sample

Line 43 – does this refer to modern or historic data, and how does this compare to communities living on islands?

Explain what is meant by ‘local island environments’ given that this study compares many different island groups

Line 59 ‘not necessarily sheep’ and ‘easily transported’ – Explain.

Line 13-23, page 16 – feels like Results rather than Discussion

The discussion at this point starts to discuss feasting and activity areas on site, without much context for understanding what happened at the site, or why this might be important. This needs to be better contextualised for the reader, as it is now, it seems out of place.

A few times on page 16 there are statements of bones being ‘biomolecularly-identified’ or caprines ‘found archaeologically’ – both of these statements are unclear and can be misleading for many readers

Line 9, page 17 – what does ‘they’ refer to here?

Line 38, page 18 – not a complete sentence

Conclusion

Line 26 – what is meant by ‘eastern Africa’s coastal tropics’ ?

Line 45 – what is the ‘Swahili World’?

Line 45– how would stable isotope analyses shed further light on these issues?

Supplementary Files

A lot of good information here, and it is nice to see such detail for all of the sites and background info for the samples.

I think it would make it much more streamlined to have Supplementary Data 3 as a landscape table within Supplementary Data 1 word document, so that the references for all of the Supplementary files can be listed in this document for ease. I think this would also help to relate the info in Supplementary Data 1 with the archaeological sites and dates. The table entitled; ‘Context Associated Caprine Dates’, for example, is a bit hard to understand. These dates are critical for the overall analyses of when these identified specimens were introduced, so the dates and the interpretation of them in relation to the specimens is key.

In the Supplementary Data 2, in the Notes column – I think it is a bit confusing to say “Missing #/#”. This supposes that you need a certain set of markers to identify, which could be misleading. It would be helpful here to either explain what this means, and say that your marker nomenclature and identification is based on Brown et al. 2020, for example, and that if x amount of markers enables an identification then explain that here so that the reader can follow your method of making these identifications. It would also be helpful to say how you peak picked the samples, against what S/N threshold.

Appendix B

Comments from Reviewers and Editors:

Response: We thank all Reviewers and Editors for taking the time to provide comments and insightful feedback for this manuscript.

Reviewer: 1

Overall, there is a lot of interesting research presented in this paper and a large amount of data contributing to the understanding of animal domestication and translocation in eastern Africa. The authors have done a commendable job of pulling together historic archaeological excavated data sets, radiocarbon dates, morphological info with a robust data set of new collagen fingerprinting analyses. Thus, there are some important insights here which have a wider impact than just presenting the archaeology of these island sites, demonstrated by the paper trying to pull together so many strands from different disciplines. The reach of the paper, however, in the way it is written, is quite narrow in terms of overly focusing on the archaeology of the islands without really broadening this out to the bigger debates on domestication in an East African context, interaction with mainland eastern Africa, as well as how different some of the islands are from each other, and how that might play in to a big-picture comparison of the movement of humans, animals, and plants across the different island groups. The text is repetitive, too long, and the language is unclear at points, with long sentences, which makes it hard for the reader to get through to the key points and important new research that has been done. For this work to reach a wide readership in a journal such as Royal Society Open Science, it needs to be much shorter and punchier, without as much specific archaeological information in the main text. Even for a readership of archaeologists working in the region and familiar with sites and the big questions, the arguments are not presented in a clear and organised fashion to follow the take-home messages and significance of the new data presented. At present, it reads much more like a thesis, and thus would benefit from a thorough editing to shorten it and distil the key points for the reader.

Response: We have reviewed the manuscript carefully to reduce long sentences, avoid unclear language, and where possible shorten the overall length. Changes are tracked in the appropriate version of the attached resubmission.

The word 'Swahili' is used differently in this paper, sometimes to denote the coast and sometimes to denote the islands as a group, and other times to talk about cultural groups and ways of life. My first point about this is that it needs defining somewhere in the introduction, especially for a broad readership who might associate the word with the language, rather than the cultural or historical meaning or the meaning here to signify a lifeway, heritage, etc. I also wonder, then, about the use of it in the paper to refer to the island groups that are compared in the study— as there are many references, especially in the discussion, to 'Swahili islands'. I wonder if this is a bit problematic to group them in this way, especially without defining the term, and clarifying if this means the time period and way of life that connected the communities living on these islands, or if it is being used as a geographical term? Give this some thought and define it as the authors wish it to be defined and used throughout the paper so that the reader is absolutely clear what is meant by the term.

Response: We thank Reviewer 1 for this comment and agree. To avoid any confusion, we have removed the term 'Swahili' completely from the manuscript.

Abstract

Lines 5-6

This sentence is quite clunky for the abstract. It does not say that this material is from archaeological sites, and the term Iron Age is separated from the chronology, which is difficult for the average reader who might not know that the Iron Age spans that time period in eastern Africa. 'main early colonisation phases' is also quite difficult to interpret

Response: We have revised the wording to indicate that the analysis was on zooarchaeological remains, clarified the Iron Age chronology, and removed the latter part of the sentence about colonisation phases. Wording has been tightened where possible.

Line 10 - 12

'caprine presence' – odd phrasing – perhaps to be as clear as possible for the abstract, just write that previous zooarch analyses identifying caprines in the remains, or in the faunal record

Response: We have removed the word 'presence' so it now reads: "Where previous zooarchaeological analyses had identified caprine remains at four of these sites..."

'this study found..' not very clear - sounds like the team found these remains whilst excavating, or they were lost, rather than identifying these species using collagen fingerprinting, so be clearer here 'or had only tentatively' – rather weak point for the abstract, which should be short and punchy and draw the reader in. Remember that some readers only read the abstract, so this should be all of the key points of the paper distilled into these sentences. Is it because the remains were fragmentary, and difficult to identify using standard morphological techniques?

Response: We have clarified and tightened the wording as suggested (along with the following sentences, see next comment). The sentence now reads: "Where previous zooarchaeological analyses had identified caprine remains at four of these sites, this study identified goat at seven sites and sheep at three, demonstrating that caprines were more widespread than previously known."

Lines 12 – 23

Too wordy for an abstract. Distil these ideas into one sentence.

Response: We have reduced these last two sentences of the abstract as suggested.

Introduction

Best to put a space between the number and the unit '3000 km'

Response: We have corrected this in the text, and also changed all other units of measurement so that there are spaces between the number and unit, as per recommended SI style conventions.

For the opening sentences of the introduction, these sentences are not very clear about what this paper is trying to introduce. Trans-continental – yet Africa is a continent and the Indian Ocean world is a rather nebulous entity? Concrete examples here would help as well as being far more plain and clear in the language about the history of exchange between eastern African islands and the Indian Ocean – due to seafaring technology, exchange of specific crops, etc.

Response: The introduction has been revised to clarify these points.

What is meant by ‘world regions’?

Response: We have revised the introduction and this phrase no longer appears.

Explain what is meant by the Swahili coast

Response: As above, the term ‘Swahili’ has been removed from the manuscript to avoid confusion. We have used ‘island (or coastal) eastern Africa’ in its place.

The last few sentences of the first paragraph of the Introduction need to be tightened, as it jumps between the work done on the Swahili coast and in a global context on island systems and therefore uses a lot of buzz words – it is then difficult to know what the authors are actually trying to say.

Response: The introduction has been shortened, including by removing the global context and focusing on the work done on the East African coast.

Lines 48-60

‘likely’ is over-used, and is unclear where the introduction of these non-native species came from – how did they get there, and from where?

Response: Our over-use of ‘likely’ has been remedied in the revised Introduction, and we have clarified that the non-native species mentioned (chicken, cat, rat, mouse) were introduced from Asia via maritime trade.

‘mid-late’ as a classifier for a date range is a bit ambiguous – explain this in numbers for the reader, especially with use of the term Iron Age here again

Response: We have removed the chronological ambiguities and simplified the sentence so it now says “...during the Iron Age (c. 100–1000 CE)”.

Lines 9 – 46

‘despite’ used twice in succession.

Response: This repetition has been rectified.

First sentence of this paragraph is quite difficult to understand that caprines are sheep and goat

Response: This sentence has been reworded to remove the word 'caprines', thus avoiding confusion and reducing verbosity. It now reads: "Less attention, however, has been paid to questions surrounding the introduction of livestock such as cattle (*Bos taurus*, *B. indicus*), goat (*Capra hircus*) and sheep (*Ovis aries*) to the islands."

'Western Indian Ocean islands' 'islands globally' – explain?

Response: We have removed the relevant part of this sentence to avoid ambiguity and verbosity.

Line 23

Which islands?

Response: This phrase/sentence no longer appears in the text.

Line 24 - 25

Have not introduced 'the Swahili' so difficult for the reader to understand what the 'Swahili subsistence economies' means

Response: As noted above, we have removed the term 'Swahili' from the paper.

What does 'regional' refer to here?

Response: This phrase/sentence no longer appears in the text.

Lines 42-43

Why is it notoriously difficult? Explain here or combine with above sentence about fragmentation and poorly preserved remains, but also how it is difficult morphologically when you don't have specific elements of the skeleton, it is compounded by them being fragmentary and poorly preserved. Explain this succinctly for a non-zooarchaeological specialist

Response: This phrase/sentence no longer appears in the text. We explain the issues surrounding the identification of archaeological caprine remains in more succinct terms elsewhere.

Line 47

I don't think collagen fingerprinting is a recent development

Response: We agree with Reviewer 1, the phrase 'recently-developed' has been removed.

Line 51

Delete 'assist in' and be assertive with the findings

Response: 'Assist in' was removed.

1.1 Island colonisation

Section 1.1 could really be tightened up and more helpful to have a comparison of the histories and ecologies of the island groups that are being compared, as they have very different human settlement histories – it would be great to either group/ organise them in this section, or have sub themes to discuss (briefly) where the humans came from and when, and what plants and animals were translocated and whether there was also influence from Indian Ocean trade. Some of this information is in the current section for some of the island groups, but not all. Because we know these histories are all quite different, based on previous work, I think it's important to emphasise this to the reader, especially as the paper groups the islands together for comparison.

Response: We have significantly revised and expanded this section (now section 2) to address all of Reviewer 1's concerns.

Line 14

Delete 'highly'

Response: 'Highly' was removed.

Line 16

This sentence could usefully be moved to the intro so that the intro covers the interaction between the islands and mainland, but still could be explained much more specifically on what is meant by 'culturally-mediated biological exchanges'. Explain who and where people came from and what they exchanged, what their lifeways and motivations were so that the reader gets a clear picture of what is meant by this phrase.

Response: We have deleted this sentence as part of our revisions, but addressed the reviewer's concerns about providing context for species introductions in the Introduction through other changes to the text.

I find Figure 1B quite hard to read and think it could be better displayed with a horizontal bar perhaps with the date ranges and a clearer comparison between the 2 methods, as it is difficult to see when the species level identifications are lumped with the more general identifications 'mammal' - it is difficult to see that the biomolecular identifications helped attain a better, more specific analysis. Wording of the figure caption is also hard to read : 'caprine related ZooMS identifications' and 'only those samples that returned a biomolecular identification' – what does this mean, in plain language?

Response: We agree that this figure is confusing, and does not add information that is not displayed elsewhere. As such, it has been removed and the caption shortened.

Line 11 - 38

‘culturally-mediated’ is used many times throughout the manuscript without much context to explain who this refers to, what was exchanged, from what directions, why, etc.

Response: As noted in response to the comment above, we have clarified the human agency involved in these biological exchanges and their relationship to local migration of farming communities and international maritime trade.

Delete ‘also’

Response: We disagree with the suggestion to delete ‘also’ from this sentence. We believe that it needs to be retained in order to highlight the recent addition of new evidence from Zanzibar: “The extinction of Madagascar’s megafauna is perhaps one of the better studied (though still widely debated) cases (e.g., 50, 51-55), but recent archaeological evidence has **also** implicated humans in the extirpation of a suite of wild fauna on Unguja (1).”.

‘recent evidence’ – recent archaeological evidence?

Response: We have changed the wording as suggested: “but recent **archaeological** evidence has also implicated humans in the extirpation of a suite of wild fauna on Unguja”

‘were present archaeologically’ – not very clear

Response: We have removed the term ‘archaeologically’ from this sentence to improve clarity.

‘late Pleistocene’ – give dates for this in brackets for the reader

Response: Dates have been added for the late Pleistocene (c. 20,000 years ago) when Unguja was connected to mainland Africa.

Sentence starting ‘Their local...’ needs to be split up

Response: We removed part of this sentence to reduce the manuscript length. It now reads: “While environmental isolation following Unguja’s separation from mainland Africa likely contributed, new hunting pressures and habitat impacts associated with the arrival of farming populations (e.g., the clearance of natural vegetation for settlement and agriculture, the introduction of commensal species such as black rat) are also thought to have had a significant impact (5, 9, 68, 69).”

Line 45

Wild mainland species? As only the bush pig is listed as ‘wild’ in this sentence

Response: We thank Reviewer 1 for picking up this error, it has been amended in text.

Line 50

‘were possibly moved’ explain this lack of certainty

Response: This has been changed to ‘introduced’, thereby removing the uncertainty: “common tenrec (*Tenrec ecaudatus*) and two species of lemur (the common brown lemur, *Eulemur fulvus* and the mongoose lemur, *Eulemur mongoz*) were **introduced** to the Comoros from Madagascar”.

Line 57

Colonised by...? Or inhabited by humans? In general, an overuse of the word ‘colonised’ in the manuscript without a context

Response: We have reviewed our use of the word colonised throughout the manuscript, though we note that it is unavoidable in many contexts where we describe the first human occupation of the islands.

Line 9

‘is lacking’ – better word choice there to end paragraph

Response: ‘is lacking’ is no longer at the end of a paragraph.

Lines 12-34

Explain why ‘Iron Age’ is in quotes

Response: We have removed the phrase in question (“especially as part of colonising ‘Iron Age’ economies”) from the text as it did not add any clarification.

Would be useful to explain here the use of domestic/wild animals on the mainland, and how this compares with island translocations

Response: We have provided brief context where relevant to the use of fauna on the mainland.

‘much later’ – quantify this if possible

Response: We have provided more precise date ranges for the arrival of goat and sheep so that the time interval is more easily discerned: “The earliest zooarchaeological evidence of goat on the islands dates from c. 700 CE (22-25, 30, 38, 44, 65, 68). Sheep, however, have only been identified in assemblages at three sites in the Zanzibar archipelago dating to c. 1000–1200 CE (20, 37), suggesting a much later introduction compared to goats”

‘it is suggested’ – based on what?

Response: The text has been modified to read: “**Zooarchaeological evidence** suggests that sheep only arrive in the Comoros at around the same time, in the 11th–12th centuries CE”

‘Early Iron Age’ – first time this is used, without Iron Age ever being contextualised

Response: These time periods are now contextualised in Section 2.

Why is there no robust evidence? Explain the lack of excavation, etc here

Response: We have explained that preservation issues may be in part to blame for this situation. Additionally, we have outlined other controversial evidence for a much earlier introduction of species in section 2.2, 'Also problematic are claims for the introduction of goat as well as other domesticates (dog, *Canis familiaris* and chicken) to the islands between 5000–3000 years ago (77, 85), which have not been backed up with robust chronological or osteological evidence (86) or replicated by subsequent studies (1, 2, 27, 31)."

Lines 36-59

'compositions' of what?

Response: This has been clarified as referring to 'herd compositions'.

Distinctions/distinctive used multiple times in one sentence

Response: We have changed 'distinctive' to 'characteristic' to avoid this repetition: "These distinctions are significant, as these two taxa are behaviourally different with **characteristic** dietary preferences and ecological impacts."

'This raises...' what is the chronology of when these events occurred, and is this the case for all of the island groups that are being discussed?

Response: The sentence in question has been deleted as part of our broader revisions.

2.1 Sites

Delete 'widescale'

Response: The sentence in question has been removed for the sake of brevity.

Replace 'antiquity' with chronology or other more precise term

Response: 'Antiquity' has been replaced with chronology.

'Early Iron Age' – date?

Response: We have provided dates for the Early and Middle Iron Ages previously in the text.

'Cave sites' sentence too long – also what were these mixed subsistence economies? I don't think by this point the reader has a clear picture of the differences between communities inhabiting the mainland and islands of east Africa in the Iron Age

Response: We have broken this sentence up to reduce its length, and added text here and previously in the manuscript to clarify the subsistence economies of different communities occupying the mainland and islands.

2.2 Sample selection

'as being as close to caprine as possible' - what does this mean? Explain

Response: We have changed the wording of this sentence, and also provided clarification earlier in the methods to explain the different bovid size classes. The main text now reads: "fragmented bones that were identified morphologically as caprine (e.g., '*Capra hircus*', 'caprine') or **less identifiable specimens that could potentially be caprine (e.g. 'Bovoid Size 1-2' and 'Bovoid Size 2')**."

'pre-date all confirmed evidence to the Swahili region' – what is this, and when? Is there a Swahili region?

Response: As noted above, we have removed the term Swahili from the paper to avoid ambiguity. This now reads: "pre-date all confirmed evidence for the arrival of domestic livestock to eastern Africa's coast and islands".

'broader categories' – this is ambiguous
'less specific taxonomic classes' – ambiguous

Response: These two comments relate to the same sentence, which we have changed as follows to clarify: "For three sites (Unguja Ukuu, Fukuchani, Juani Primary School), we also sampled a random selection of highly fragmented specimens that could only be identified to **broader taxonomic levels** (e.g., 'mammal' or 'vertebrate'), in order to assess the usefulness of ZooMS in identifying domesticates among poorly preserved bones.

Only in the final sentence of this section does the reader know that ZooMS analysis was used – move this earlier.

Response: We added "for ZooMS analysis" to the first sentence of Section 3.1 (Sites), and changed the third sentence of Section 3.2 (Sample Selection) to read: "Given the main aim of the present study was to use ZooMS to test for the presence of caprines...", so that the reader is reminded that ZooMS analysis is being used.

2.3 ZooMS protocol

For correct use of 'stable isotope' to describe the analyses say that the bone collagen was also extracted for stable isotope analyses, and that the preparation method to extract collagen for stable isotope ratios or measurements instead of analysing for 'stable isotopes' or 'stable isotopes preparation method' . See Chapter 2 of Sharp's book for more help on this: https://digitalrepository.unm.edu/unm_oer/1/

Response: The wording in what is now Section 3.3 has been changed to reflect this recommendation.

It is important to explain why you must avoid diagnostic features for future research/ comparing previous analysis

Response: This sentence has been modified to: "...whilst avoiding any morphologically diagnostic features (e.g., epiphyses), to preserve the specimens for future morphological and metric analyses."

It is also crucial to explain why you use 2 different established ZooMS protocols, as they were developed for specific purposes. The wording here is also a bit strange, as the extraction methods are not necessarily 'accessing' or 'utilising', as the gelatinisation and solubilising during acid demineralisation are not exactly correct here, if strictly following the protocols in the references listed. Explain what buffers you used and at what PH for the extractions, and why it was necessary to follow 2 different methods. These are important to detail, as I also couldn't find a direct comparison in the results and discussion section about the use of the 2 methods. This is important, as one of the issues the results raised was the preservation of collagen from the material and the ability to extract enough collagen for identification. These 2 methods have a big impact on that, as the HCl (depending on molarity, volume, how many times it is changed) will be more aggressive and if not watched carefully and in samples with very little collagen yield already, could destroy the organic. For the readership of Royal Society Open Science, and for archaeologists working on poorly preserved material trying to use these methods, the comparison of these extractions on this large data set is important, and should be highlighted in the Results and Methods, and perhaps include a table or graph in the supplementary showing which method was best, etc.

Response: We have re-written this section to address Reviewer 1's various concerns, and to highlight our desire to be as economical as possible in our approach to analysing our samples by using both methods for simultaneous extraction of collagen in the event that one gave better results. We have also added a section to the results discussing the efficacy of the different protocols (where the acid insoluble method proved to be superior), and Supplementary Data 2 now indicates which protocol was used for every sample.

In the Zenodo files, I could only open the spectra in mMass and not necessarily see which extraction method was used for the averaged spectra. I would put the files on Zenodo as text files as well, simply because text files can be imported into any mass spec software programme.

Response: Spectra have now been uploaded as txt files and can be found in [10.5281/zenodo.4767341](https://zenodo.org/record/4767341/files/10.5281/zenodo.4767341).

Lines 45 – 56

Need more detail here, specifically about the machine settings, plate used, etc. Refer to the references here and their Methods sections for the details typically added in this section for ZooMS analysis.

Response: This information has now been included.

Results

I find the first paragraph of the results section difficult to read, as it is essentially a word description of Table 1. Try to summarise all of the results more succinctly, rather than choosing some sites to give detail of the success rate, or phrases such as 'double the success rate' 'very low success rate – worst of all the sites' as this is not really helpful to the reader for understanding a clear report of the results, and doesn't mean much outside of the comparison of this data set. I also think that in cases where there are such different sample sizes (1/8) it is actually a little misleading to try and frame the results as percentages against one another. As this section rightly says, the variations are probably due to a range of different factors, and in the last sentence, it would be helpful to reference the wealth of literature on factors affecting collagen preservation.

Response: We agree with Reviewer 1. We have provided a more succinct summary of the results and preservation rates, and key references were added for collagen preservation issues.

Line 28 - what is defined as a 'successful' sample? 'species-specific identification' clunky

Response: We removed these two phrases so it now reads: "The majority of the samples identified using ZooMS were *C. hircus* (n=259/318, 81%) (Table 1). This species was identified at seven of the eight sites investigated (all except Juani Primary School), including for the first time at Makangale Cave, Ukunju Cave and Sima."

Line 33 – *hircus* in italics

Response: We thank Reviewer 1 for picking up on this error, *hircus* is now in italics.

Line 48 – Probable? Explain this.

Response: We have replaced 'probable' with 'other' so it now reads: "Other domestic or commensal species identified....".

Line 60 – via ancient DNA? Explain this.

Response: We have changed this to: "via ancient DNA analysis".

Lines 5-40 on page 10: This paragraph is difficult to read and make sense of, and I wonder if it would be better to be either supplemented or shortened alongside a table instead. Also, some statements need to be explained, such as 'missing too many markers for species identifications'. It is important to have a table in the main text of the paper with the ZooMS markers which were used to identify caprines at the sites, and perhaps a list of the markers which were used to identify the wild species and a simple list with markers of the wild species that were found. As it is, Figure 2 and 3 are helpful to get an overall view, but there is no ZooMS marker data in the paper as a table at present. I realise that there were a lot of samples analysed, so the full data set needs to be in supplementary, but it is important to highlight for non-specialists how the reference markers/spectra were used to make the

identifications. This is especially important given that many of the spectra are reported as compared against a data set in a paper that is submitted, but not in press (Janzen et al. submitted).

Response: We have shortened this paragraph significantly by providing only a brief textual summary of the results and referring instead to Table 3 for details. We have also revised and shortened Table 3, moved the original table to the Supplementary Data and presented a simplified summary of the non-caprine fauna in the main text. We have also added a table (Table 2) to the main text listing the reference ZooMS markers used to identify taxa in this study. We also note for the reviewer's reference that **Janzen et al. has now been published** (<https://journals.plos.org/plosone/article/comments?id=10.1371/journal.pone.0251061>).

For the samples that had good spectra, but no identification was possible, it would be useful to explain in the Results section whether the authors tried to match against existing ZooMS data, possible contaminants, or whether the exercise was simply to identify caprines and if not, then it was discounted as 'unknown'

Response: We have added the following statement to the ZooMS protocols to clarify: "For samples containing high amounts of collagen but where no identification was possible comparing the peaks to the reference library, these 'unknowns' were included in case of future improvements to the reference dataset."

Lines 43-60 - this paragraph is also difficult to read, and could be summarised with a table. Phrases such as 'remained at the caprine level' are unclear. Also, not sure the average reader will understand the various zooarchaeological size classes of "Bovid" or "Mammal Size Class" so this needs to be explained and contextualised if used here. Same problem with caption for Figure 2.

Response: We agree with Reviewer 1 that this paragraph is difficult to comprehend. Given that Figure 2 summarises this information graphically, we have deleted this paragraph. We have also clarified the various size classes in Section 2.2 of the Methods and added a table of examples to Supplementary Data 1. The methods text now reads: "Due to high rates of fragmentation, bones were often only able to be identified morphologically to broader taxonomic categories (e.g., 'vertebrate', 'mammal' and 'bovid') and **body size class (adapting 83 to eastern African bovids). For example 'Bovid Size 1' is the size of dik-dik or suni, 'Bovid Size 1-2' is the size of bush duiker, klipspringer, oribi, or small caprine, and 'Bovid Size 2' is the size of caprine, bushbuck, or Thomson's gazelle; see also Table S1.1 in Supplementary Data 1)**".

Line 52 - needs a reference here, or explanation of how this is known without LC-MS/MS data

Response: We have added citations to published data for the identification of marine mammals to the reference library tables (Table 2 and Supplementary Data 2).

Discussion

For me, the first paragraph and Figure 3 in the Discussion are really key points of the paper, and need to be featured in the abstract and intro as the significant outcomes of the work. Figure 3 really sums up the research nicely and I wonder if incorporating a graphic like this one for Figure 1b would be helpful.

Response: We thank Reviewer 1 for these positive comments. We have now flagged the study's most significant findings regarding the chronology of caprine introductions in the abstract and also the introduction. We prefer, however, to retain Figure 3 in its current location — Figure 1b has been deleted and Figure 1 is now just a map showing the location of sites mentioned in the text, which seems appropriate as a standalone figure.

The first paragraph on page 13 has a lot of unclear phrasing 'earliest reliably-dated ZooMS-identified goat occurs' ' here it was found to be present' ' goat first appears in context with a potentially wider timeframe' I'd recommend re-writing this paragraph using clear, succinct language.

Response: We have revised the wording to be clearer and more succinct where possible.

Line 35 – explain what you mean by 'lowermost contexts' for a non-specialist

Response: We have clarified that we mean the lowermost excavated layers of the site, which are the oldest.

Line 42 – based on the data presented –not usual to start a new paragraph with 'on the other hand' as it should follow from thought directly before

Response: We have removed this text.

Line 45 – 'before now' – *prior to the results presented here?*

Response: 'before now' was changed to 'before this study'.

Line 51 – 'ZooMS-confirmed sheep occurred' ?

Response: This wording was changed in the manuscript.

Line 53 – 57 explain what this means to someone who is not an archaeologist reading this

Response: We have removed reference in the text to Dembeni phase pottery, stating instead that the caprines appear "in association with 8th-10th century pottery".

Line 26, page 14 – 'well-dated' ' site's size and wealth' – what does this mean – explain this here and give specific details of why these observations are important to the argument

Response: We agree with Reviewer 1 that clarification on all these points was needed. These sentences now read: “The comparatively early presence of caprines at Unguja Ukuu most likely relates to sampling bias (the site had the largest assemblage analysed in this study and also has the most robust radiocarbon chronology, allowing more precise time frames for caprine arrivals to be established) as well as its larger size and wealth as a major Indian Ocean trading hub compared to other sites in the study. The latter probably afforded its occupants better access to domesticates than more rural localities such as neighbouring Fukuchani.”

Line 38 – ‘ZooMS-identified caprines’ unclear

Response: We have removed ‘ZooMS-identified’ from this sentence, so it now reads: “The absence of caprines in the Early Iron Age assemblage from the Juani Primary School site...”.

Line 55 – ‘It nonetheless highlights’ ... this sentence is key and a really important point, but is couched in between wordy sentences, so would suggest bringing this out and contextualising it as a key point that this paper makes in light of the new data set presented

Response: We have shortened the preceding sentences so that this point now stands out.

Line 7, page 15 – ‘maritime-oriented societies’ – what does this mean? Explain.

Response: This phrase has been removed as part of our revisions to shorten the manuscript.

Line 21 – much greater importance or just now a much greater number? Explain.

Response: We have clarified our wording to support the argument that goats were economically more important - they are present in much greater number overall, they are more ubiquitous (present at more sites), and appear earlier on the islands than sheep.

Line 36 – lower numbers in your studied sample

Response: This sentence was changed to: “...they are also much rarer in zooarchaeological assemblages”

Line 43 – does this refer to modern or historic data, and how does this compare to communities living on islands?

Response: We have deleted the ethnographic comparison to the Samburu, given they are geographically and culturally distant from our study region. Unfortunately, there is no relevant ethnography from the islands to draw on to help furnish our interpretations in this respect, so we are restricted from making comparisons to communities living on islands today.

Explain what is meant by ‘local island environments’ given that this study compares many different island groups

Response: We have removed 'local' from this sentence so that it is a broad statement only about the suitability of sheep for "herding in the island's environments, which tend to be dominated by forests and coral thicket scrub rather than pastures suitable for grazing".

Line 59 'not necessarily sheep' and 'easily transported' – Explain.

Response: We have removed 'not necessarily sheep' and clarified the sentence so it now reads: "make them easier to transport between the islands".

Line 13-23, page 16 – feels like Results rather than Discussion

Response: The paragraph in question, discussing temporal patterns at the site of Unguja Ukuu, has been deleted for the sake of brevity (see Reviewer 1's next comment).

The discussion at this point starts to discuss feasting and activity areas on site, without much context for understanding what happened at the site, or why this might be important. This needs to be better contextualised for the reader, as it is now, it seems out of place.

Response: As noted above, we have deleted this paragraph on temporal trends at Unguja Ukuu. We agree that the discussion was speculative and needed more site-specific context, which would have been out of place in the present paper.

A few times on page 16 there are statements of bones being 'biomolecularly-identified' or caprines 'found archaeologically' – both of these statements are unclear and can be misleading for many readers

Response: As above, the paragraph where these phrases occur has been deleted. We have also reviewed the rest of the text to ensure similar phrases are clarified.

Line 9, page 17 – what does 'they' refer to here?

Response: This paragraph was edited for clarity as suggested.

Line 38, page 18 – not a complete sentence

Response: We agree, and have fixed this sentence accordingly.

Conclusion

Line 26 – what is meant by 'eastern Africa's coastal tropics' ?

Response: This sentence was changed to describe how difficult it can be to get readable collagen from a tropical environment.

Line 45 – what is the 'Swahili World'?

Response: As noted above, we have deleted all use of the term Swahili, the sentence now reads: “...to help understand the potential long-term ecological impacts of faunal translocations on eastern Africa’s island environments...”.

Line 45– how would stable isotope analyses shed further light on these issues?

Response: this example was given more context for the reader (“to detect preferences for feeding habitats”)

Supplementary Files

A lot of good information here, and it is nice to see such detail for all of the sites and background info for the samples.

I think it would make it much more streamlined to have Supplementary Data 3 as a landscape table within Supplementary Data 1 word document, so that the references for all of the Supplementary files can be listed in this document for ease. I think this would also help to relate the info in Supplementary Data 1 with the archaeological sites and dates. The table entitled; Context Associated Caprine Dates’, for example, is a bit hard to understand. These dates are critical for the overall analyses of when these identified specimens were introduced, so the dates and the interpretation of them in relation to the specimens is key.

Response: We agree with Reviewer 1 and have moved Supplementary Data 3 to the Supplementary Data 1 document. We have also reviewed the “Context Associated Caprine Dates” table so that the chronological information is explained more clearly for a non-archaeological reader, and better highlights the earliest dated occurrence of sheep and goat on the islands.

In the Supplementary Data 2, in the Notes column – I think it is a bit confusing to say “Missing #/#”. This supposes that you need a certain set of markers to identify, which could be misleading. It would be helpful here to either explain what this means, and say that your marker nomenclature and identification is based on Brown et al. 2020, for example, and that if x amount of markers enables an identification then explain that here so that the reader can follow your method of making these identifications. It would also be helpful to say how you peak picked the samples, against what S/N threshold.

Response: We thank Reviewer 1 for pointing out these issues. We have removed the ‘missing markers’ column from the notes in Supplementary Data 2, which we agree is unnecessary for readers. We have also added in the threshold (S/N) for identifications into the methods section of the paper.

Reviewer: 2

There are some minor changes needed and some grammatical corrections, otherwise this is an excellent paper.

We thank Reviewer 2 for their comments, and are pleased to hear that they thought our paper was excellent. We have taken their suggestions on board as outlined below.

Page 2 (in the text) line 59 and ff. - It would be worth defining the term Iron Age here and what it is conventionally understood to imply - especially as you use the term frequently in the paper. The use of Early Farming Communities might be better - but much depends on whether you are using the term Iron Age simply as a chronological marker, or whether you also mean to imply inception of farming

Response: We agree with Reviewer 2 that clarification was needed. We use Iron Age as a chronological marker only so we have added a date range to the Introduction. We have removed any usage of the term to describe cultural associations such as (for example) “Iron Age peoples/groups/communities”.

Page 3, lines 21-22 - for the benefit of those unfamiliar with the regional archaeological terminology and chrono-stratigraphy, it would be helpful if you could point out that domestic stock had been present in mainland East Africa for much longer associated with Pastoral Neolithic (PN) assemblages - and even in the coastal hinterland by c 3500 BP (Wright 2005 on Tsavo sites, although he does not provide any morphological identifications, just sized classed inferences owing to the presence of PN ceramics).

Response: A sentence was added into section 2.2 to briefly address the presence of caprines on the mainland for ~5000 years and their importance.

Page 5, line 53 - I prefer Late Stone Age rather than Later Stone Age, as it is grammatically more correct (Early, Middle and Late). 'Later' has become the default in southern African archaeology, but there are still East Africanists who use 'Late' - and this was more common in earlier decades. Please consider changing.

Response: We appreciate Reviewer 2's comment about terminology, but we prefer to retain the term 'Later Stone Age', which has been used in the vast majority of recent publications in eastern Africa including those cited in our paper.

Page 7 (Methods, section 2.1) Lines 16-17 - I'd like to see some acknowledgement that all of these sites had been located prior to the Sealinks work - important though the contributions of that project have been. How about this phrasing here (or something akin to it): 'Located during previous archaeological campaigns led by different local and foreign researchers, these were re-visited and re-excavated between ...'

Response: Reviewer 2 makes an excellent point. We have acknowledged these earlier campaigns in Supplementary Data 1 where detailed site backgrounds are provided. During our revisions of the main text, however, we felt it best to remove the sentence in question for brevity.

Same page line 32 - stone structures were also present at Unguja Ukuu - see Fitton, T. and Wynne-Jones, S., 2017. Understanding the layout of early coastal settlement at Unguja Ukuu, Zanzibar. *Antiquity*, 91(359), pp.1268-1284.

Perhaps acknowledge that Chami exposed structural evidence for houses at Juani Primary School Chami, F.A. 2000. Further archaeological research on Mafia Island. *Azania* 35: 212–

13, and maybe also make reference to more recent geoarchaeological work aimed at detecting house floors (can you not use 'hut' please! refer to them as daub or mud-and-wattle houses) Sulas, F., Kristiansen, S.M. and Wynne-Jones, S., 2019. Soil geochemistry, phytoliths and artefacts from an early Swahili daub house, Unguja Ukuu, Zanzibar. *Journal of Archaeological Science*, 103, pp.32-45.

Response: We have changed the text to include the preferred terminology, 'daub houses' and added references to Fitton and Wynne-Jones 2017, and Sulas et al. 2019. Additional details were also added to the Supplementary Data 1.

Page 8 line 3 - replace 'were' with 'was' (a random selection ... was)

Response: This change was made.

Page 12, Fig 3 caption, line 54 - Capital Island for 'Unguja Island' for consistency elsewhere in the paper

Response: This change was made.

Page 13 - again acknowledging the very significant contributions the Sealinks project made to clarifying the settlement and occupation history of Kuumbi Cave, the suggested early presence of SE domesticates and a 'Neolithic' occupation had already been challenged before this phase of work began - especially by Sinclair - perhaps cite his 2007 paper on line 24 after 'finding,' while retaining citations to references 24 & 38 at the end of the sentence. P.J.J. 2007. "What is the archaeological evidence for external trading contacts on the East African coast in the first millennium BC." In *Natural Resources and Cultural Connections of the Red Sea*, edited by J. Starkey, P. Starkey and T.J. Wilkinson, 1–8. Oxford: Archaeopress.

Response: We agree with Reviewer 2 that this was an oversight. Sinclair's 2007 paper has been cited as suggested.

Page 14, line 7 - reference to UU being one of the oldest trading ports on the Swahili coast merits a reference to the ongoing work by Wynne-Jones, Sulas and Fitton at the site -
Wynne-Jones, Sulas & Fitton
Wynne-Jones, S., Sulas, F., Out, W.A., Kristiansen, S.M., Fitton, T., Ali, A.K. and Olsen, J., 2021. Urban Chronology at a Human Scale on the Coast of East Africa in the 1st Millennium ad. *Journal of Field Archaeology*, 46(1), pp.21-35.

Response: We thank Reviewer 2 for recommending this recent paper. It has been added to the discussion of Unguja Ukuu as a large trading port.

Page 16 - line 52 - an example of a site where this future work on later 2nd millennium faunal assemblages could be carried out is Kua on Juani Island - Christie's work hints at a growing presence of caprines, but again no formal IDs, just size-classed categorisations - see Christie, A.C., 2013. Exploring the social context of maritime exploitation in Tanzania between the 14th-18th c. AD: recent research from the Mafia Archipelago. *Prehistoric marine resource use in the Indo-Pacific regions*, pp.97-122. Perhaps more information on

taxa is available in her 2011 PhD - not sure it is online however.

Response: This is an excellent suggestion, it will be kept in mind for future research. We have opted not to mention this in the paper, however, for the sake of brevity as per Reviewer 1's comments.

Page 17 line 7 - change 'subside' to 'subsist' - neither the Cambridge online dictionary nor the online Merriam-Webster use the word 'subside' in the manner implied here - although I have seen it used in this way in other publications.

Response: This has been changed.

Reviewer: 3

Comments to the Author(s)

Review for the manuscript « Collagen fingerprinting traces the introduction of caprines to island eastern Africa”
by Culley et al.

I found the manuscript of Culley and collaborators of particular interest since it focuses on a relatively unexplored area of research: the introduction of domesticates (especially caprines) to islands of eastern Africa. The manuscript describes the application of ZooMS to various archaeological sites aiming at refining morphological identifications of faunal remains. They identified a large number of caprine remains, of which the majority is goat. They discuss the newly identified domesticates site by site and the influence caprines might have had on the islands' ecologies. Although I found the paper of great interest, I raise in the subsequent paragraphs several points that, according to me, should be improved prior to publication.

Comments:

Generally, when referring to the two caprines species, the term “caprines” is plural. Otherwise, you only refer to goat (by opposition to ovine for sheep). I suggest you check the entire manuscript to avoid misunderstandings.

Response: We have reviewed the manuscript carefully to ensure that we employ correct use of the plural “caprines” to refer to both sheep and goat, or the singular “caprine” when we are referring specifically to just one of the two species. To clarify, caprine is the common/English form of the Latin name *Caprini*, a tribe that includes both sheep and goat. Therefore, caprine refers to both sheep and goat, not only goat as the Reviewer seems to be suggesting. For the benefit of non-specialist readers, we have also clarified in the Abstract that the term refers to both species: “...the introduction of domesticated **caprines (sheep and goat)** to these islands”.

With regard to pluralisation, for example in the introduction where we say “Tracing caprine dispersals has been hindered ...”, we are using “caprine” as an adjective to modify “dispersals.” The same as we could say “alcelaphine diet,” “tragelaphine horns”, etc. It is therefore acceptable to use it in the singular. We can also use it in the singular as a noun,

e.g., “we found one caprine”. This is a standard English/common form for the Latin term Caprini. It is true it doesn’t appear in standard dictionaries as a noun, but this is a common usage.

<https://www.merriam-webster.com/dictionary/caprine>

<https://en.wiktionary.org/wiki/caprine#:~:text=hircine-,Noun,member%20of%20the%20tribe%20Caprini.>

You make a number of statements that, in my opinion, should be more nuanced and if not, more referenced. For instance, l. 43, you state that sheep and goat distinction is “notoriously difficult” from caprines assemblages. A lot of papers proposed criteria for sheep/goat differentiation (Gillis, Chaix & Vigne, 2011; Salvagno & Albarella 2017 for instance) and you should nuance this sentence by adding something regarding African environments, in which this is particularly the case (and some others).

Response: We thank Reviewer 3 for this comment. We have added appropriate references for caprine morphological criteria (Balasse and Ambrose 2005, Zeder and Lapham 2010, Zeder and Pilaar 2010, Salvagno and Albarella 2017) and clarified that the issue is exacerbated in eastern Africa because of poor morphological preservation.

I found the introduction a bit brief since the context is only quickly described and the method (which, although not broad new, is still developing) is not defined. I also would like a sentence or two describing what you imply when you describe island eastern Africa as “ecologically rich and diverse region” (p. 6, l. 5).

Response: We have provided more context to Section 2, as requested by Reviewer 3, including a brief description of the method.

I would also like to know what you mean by “morphological identification”. Do you refer to comparative anatomy only, to geometric morphometrics or a combination of both?

Response: As noted above, we have clarified this in Section 2.2: “The sites analysed in this study have over 6000 individually catalogued bones from Iron Age contexts that, prior to this research, were **morphologically identified (i.e. on the basis of comparative anatomy, using locally relevant reference collections)** to different degrees of taxonomic specificity.”.

I understood that you used published markers for your species identifications. However, I am highly upset by the fact you mainly refer to an under-review paper for many species ID, which is by essence, not yet accessible. In addition, you did not consider a COL1A1 marker for sheep/goat distinction in all your data analysis, which appears problematic since it has been highlighted that the “classic” COL1A2 marker is shared between sheep and a large number of African wild bovids and the COL1A1 marker between goat and wild bovids (Le Meillour et al. 2020). I suggest you re-perform your species identification including the COL1A1 marker, to make sure what you identified are indeed (for most of the faunal remains) domestic goat, which I am certain will be the case.

Response: We thank the reviewer for their thorough consideration of our methods. As noted previously in our replies, we are pleased to report that the **Janzen et al. manuscript**

has now been published

(<https://journals.plos.org/plosone/article/comments?id=10.1371/journal.pone.0251061>). We nonetheless provide the relevant ZooMS reference markers from their study in our Supplementary Data 2 so that these data are readily accessible.

The COL1a1 marker reported by Le Meillour et al. 2020 has been shown to be useful for taxonomic differentiation between sheep, goat and wild bovids in LC-MS/MS, but further research to identify if these markers were useful for MALDI TOF MS or ZooMS identification were not carried out as part of that study. The Janzen et al. study, on the other hand, sought to use MALDI TOF MS to discriminate between African bovids and other published ZooMS reference markers (as per Supplementary Data 2). The published collagen sequences from Le Meillour et al. 2020 were used in their study, alongside their newly generated data. Janzen et al. were able to determine that the COL1a1 peptide reported in Le Meillour et al. was not visible in MALDI TOF MS spectra and could not be used for ZooMS taxonomic identification. The Janzen et al. study was, however, able to identify other collagen peptides which could be used to discriminate between African bovid groups; and the COL1a2 marker used to differentiate sheep and other wild bovids (3033 m/z) and goat (3093 m/z) is considered highly reliable.

In your discussion, you mention 14C dates. But you do not explain how you obtained them. Was the radiocarbon dating performed on the remains you identified as domesticated caprines or are they related to the burial context (i.e., they were performed on material recovered from the same layer, etc.)?

Response: The first sentence of the Discussion provides this information: “The broad chronology of caprine introductions to eastern Africa’s islands can now be reconstructed, **drawing on associated radiocarbon dates and/or ceramic chronologies (Table 1 and Table S1.2 and S1.3 in Supplementary Data 1)**”. We have also added an explanatory note to the revised table “Context Associated Caprine Dates” in Supplementary Data 1 to explain how the various dates are obtained (e.g., bracketed by C14 dates, associated C14 dates, etc.), so that this is clearer for a non-archaeological reader.

I do have a problem with your interpretations of herd compositions. I do not criticize the outcomes; but the lack of nuance and most of all, the absence of NMI representation. I understood that you identified mostly goats in your assemblages using ZooMS. However, one bone does not mean one animal. I suggest you rephrase this part taking into consideration the NMI for each site.

Response: We agree with the reviewer that MNI is a more accurate method of calculating the abundance of goat vs sheep at our sites and making comparative statements about each taxon. We have provided a new table in Supplementary Data 1 (“Table SD1.4 Minimum Number of Individual (MNI) Goat and Sheep per Site”) and referred to MNIs in the relevant places in text.

Since only a few research teams use ZooMS or palaeoproteomics in African archaeological context, it would be interesting to include in your conclusion a bit of hindsight regarding biomolecular studies concerning caprines herding in Africa. Maybe adding a few sentences

in your introduction and including your results into the “larger picture” in the discussion or conclusion would help readers to acknowledge your study.

Response: We appreciate Reviewer 3’s suggestion but have opted not to expand on this further, in order to keep the manuscript at an appropriate length.

Finally, I suggest you re-read carefully your text since your sentences are sometimes difficult to follow (mostly due to their length).

Response: We thank Reviewer 3 for this insight, the text has been edited to reduce long sentences.

p. 5, l. 46: I am not sure that we can still use the term “recently-developed” when referring to ZooMS since the method is employed in archaeology for more than a decade now.

Response: As was also the suggestion of Reviewer 1, ‘recently-developed’ has been removed from the sentence describing ZooMS.

Fig. 1: I am not sure that the right place for this figure is in the introduction, since it appears part b presents some of your results.

Response: We thank Reviewer 3 for this observation. We have removed Figure 1b from the paper and retained the Figure 1(a) map in the Introduction.

p. 8, § 2: why do specify that sheep were identified “morphologically” after you describe the appearance of goats in the previous sentence? Where goats identified using other methods?

Response: This was clarified in the text.

p. 9, Sites: I would place Fig. 1 here to locate the sites you are referring to in the text.

Response: As noted above, we prefer to have Figure 1 in the Introduction to provide the reader with immediate context for the study region and localities mentioned in the text.

p. 10, ZooMS protocol: did you perform MS/MS analyses of your samples? Since you are working on close-related species of bovids and that the presence of wild bovids is highly possible, MS/MS is a strong method to avoid misidentifications.

Response: We did not perform MS/MS on any of the samples in the current study. However, Janzen et al. 2021 performed MS/MS on 20 bovid tribes. We are therefore confident in our identifications.

p. 11, l. 33: when mentioned once, you can refer to the species only by *C. hircus*, *O. aries*, etc. You forgot to put “hircus” in italic in the sentence.

Response: We thank Reviewer 3 for picking up on this error, it has been changed and all subsequent mentions of *Capra* and *Ovis* have been initialised.

p. 12, l. 10: Which species of Alcelaphini tribe did you use in your referential?

Response: We have updated the Reference Library of Supplementary Data 2 to include this information. Blue wildebeest, hartebeest, tobi/tsessebe, hirola were used as references from this taxonomic group. Due to their close genetic relationship, however, they cannot be distinguished above tribe level.

Fig. 2 is unclear and difficult to read. I suggest you present the information differently for clarification purposes.

Response: We thank Reviewer 3 for this comment, however, all co-authors as well as the other two Reviewers found the Figure to be legible and to clearly demonstrate the relationship between morphological and ZooMS identifications. We prefer to retain the figure for these reasons.

p. 14, l. 12: what taphonomic factors are you referring to?

Response: We have modified the text here to say: "...and taphonomic factors affecting collagen preservation at the site."

p. 14, l. 22-23: what do you mean by "confident morphological identifications"?

Response: We have removed this sentence from the text as it repeats information already discussed in the previous section.

Fig. 3: I suggest you rephrase the caption which is a bit difficult to follow.

Response: We have simplified the wording of Figure 3 caption to improve clarity.

p. 15, l. 3: "the earliest reliably-dated ZooMS-identified goat" formulation is unclear. Do you mean that you dated one of the specimens that presented spectra of domestic goat? Concerning radiocarbon dates, see my comment above. Generally, for your dating paragraph, I suggest you re-order your ideas differently since it is hard to follow and mixing cal. BP, CE and BCE dates.

Response: We thank Reviewer 3 for this comment, the dating section of the discussion has been edited and is now more clear. We have also reviewed our dating terminology to be more consistent throughout the paper.

Same comments for the following paragraphs on sheep.

Response: This text has been amended.